# Preclinical MRI Using Hyperpolarized ^129^Xe

**DOI:** 10.3390/molecules27238338

**Published:** 2022-11-29

**Authors:** Stephen Kadlecek, Yonni Friedlander, Rohan S. Virgincar

**Affiliations:** 1Department of Radiology, University of Pennsylvania, Philadelphia, PA 19104, USA; 2Firestone Institute for Respiratory Health, St. Joseph’s Healthcare Hamilton, Hamilton, ON L8N 4A6, Canada; 3Department of Translational Imaging, Genentech Inc., South San Francisco, CA 94080, USA

**Keywords:** hyperpolarized, xenon, ^129^Xe, HXe, preclinical, pulmonary

## Abstract

Although critical for development of novel therapies, understanding altered lung function in disease models is challenging because the transport and diffusion of gases over short distances, on which proper function relies, is not readily visualized. In this review we summarize progress introducing hyperpolarized ^129^Xe imaging as a method to follow these processes in vivo. The work is organized in sections highlighting methods to observe the gas replacement effects of breathing (Gas Dynamics during the Breathing Cycle) and gas diffusion throughout the parenchymal airspaces (3). We then describe the spectral signatures indicative of gas dissolution and uptake (4), and how these features can be used to follow the gas as it enters the tissue and capillary bed, is taken up by hemoglobin in the red blood cells (5), re-enters the gas phase prior to exhalation (6), or is carried via the vasculature to other organs and body structures (7). We conclude with a discussion of practical imaging and spectroscopy techniques that deliver quantifiable metrics despite the small size, rapid motion and decay of signal and coherence characteristic of the magnetically inhomogeneous lung in preclinical models (8).

## 1. Introduction

The lung is inherently a dynamic system [1,2]. In the healthy organ a variety of processes across a wide range of time-scales must operate in tandem to support the oxygen/CO_2_ exchange requirements of the entire organism. At the same time, the healthy lung capillary bed has access to the entire blood supply during each pass through the circulatory system, and therefore participates in systemic metabolic processes [3,4] that require this rapid access.

Lung disease and its diagnosis is therefore extremely challenging, requiring an understanding of the muscular and airflow aspects of breathing on the scale of seconds and centimeters, to the diffusive dynamics of gases in the sub-millimeter regime, down to the millisecond- and micron-scale transport of gases across capillary walls and in red blood cells. While disease often manifests as gross structural changes visible to X-ray based imaging [5,6], or as airflow limitation measurable at the mouth [7,8], early disease, or the root cause of established pathology, is not easily discernible using these tests [9,10].

It is therefore useful to investigate the dynamics at all spatial and temporal scales, and in particular the dynamics of gas transport between various functional compartments within the organ. Among imaging modalities, MRI is particularly well suited to this task because it derives from and can make use of techniques that encode rapid dynamical behavior as changes in the frequency, linewidth or intensity of spectral features [11,12,13,14]. In the case of hyperpolarized ^129^Xe MRI (hereafter HXe MRI), the highly deformable xenon atom is subject to easily quantifiable changes in frequency depending on its location in the gas phase, dissolved in lung tissue, bound to hemoglobin inside red blood cells, and throughout the body as it leaves the blood and dissolves into various other organs and structures, in analogy with systemic oxygen transport. Established NMR methods, specialized for the challenges of an in vivo system which is always in motion and magnetically inhomogeneous, are then used to observe the life-sustaining transport and diffusion of gases along the entire path from inhalation to hemoglobin binding and subsequent transport, or to exhalation.

This review is structured to follow that path as closely as possible. Section 2 describes imaging methods to observe ventilation, the flow of gas down branching airway tree and into the acinar structures. Defects in ventilation lead to inefficient replacement of gas during the breathing cycle, loss of air/blood gas exchange capacity, and potentially poor blood oxygenation [15]. Section 3 follows the gas to the smallest parenchymal structures in which gas transport via diffusion predominates, and highlights microstructural abnormalities that can reflect tissue loss in emphysema or changes arising from alveolar collapse.

In Section 4, we introduce additional spectral features related to xenon dissolution into lung tissue and, depending on species and acquisition strategy, binding to hemoglobin and transport to distal tissues. Comparison of the rates and locations of transitions among these spectrally distinct compartments to physical models allows us to infer details about functional units which are otherwise inaccessible to any in vivo imaging technique-- either because of the microscopic size or the lack of a physical boundary between the relevant structures. Progress refining these models and using them in particular to quantify the microscopic motion of gas within the parenchymal tissue structures is summarized in Section 5.

The final conceptual steps in the dissolved xenon atom’s journey are either the return to the gas phase within the alveolus prior to exhalation, or delivery via the vasculature to other perfused organs or structures. In the former case, contrast related to lung function can be imposed by magnetically tagging dissolved atoms and reading out the effect on the gas signal. A generalization of Chemical Exchange Saturation Transfer (CEST) methods applicable to exchangeable protons in traditional NMR and MRI [16,17,18], this method is termed ‘Xenon polarization Transfer Contrast’ (XTC) MRI and is discussed in detail in Section 6. For HXe atoms carried further downstream diffusion into other tissue types leads to additional spectral features that depend on the type, perfusion, oxygenation and perhaps pathology of the tissue. Section 7 details early steps in interpreting and utilizing these features.

As the measurements become further removed from direct imaging of the inhaled gas, interpretation must be done with caution and only after proposed contrast techniques have been standardized, relevant physical variables populating the model measured, potential imaging artifacts understood and minimized, and comparisons pursued using relevant disease models in which histological or biological validation is practical. With few exceptions, these opportunities are only available in preclinical disease models; their use is therefore necessary to understand and utilize the new contrast mechanism available with HXe imaging. However, particular care must also be taken to ensure that animal models recapitulate the central features of human disease. Further, anatomical and body size differences impose different constraints on preclinical imaging than we are accustomed to in the clinic-- breathing, heart rate and circulatory rates in rodent models are far higher, as are the available imaging magnetic fields and gradients, and dedicated coils and gas delivery systems are required. Tidal breathing volumes are small, limiting the available signal, but on the other hand available gas doses are such that signal averaging over many breaths is practical.

Here we focus on qualitatively different imaging techniques that are optimized for the preclinical milieu, while maintaining the critical correspondence to methods and diseases with clinical relevance. Notably, techniques and biological contrast mechanisms are often derived from and applicable to any other MR imaging scenario in which the nucleus of interest experiences frequency shift in as it transitions among biologically relevant compartments. This includes ^1^H metabolite or fat/water imaging, or hyperpolarized ^13^C metabolic imaging. However, these standard methods often benefit from modification to accommodate the large magnetic inhomogeneity of lung tissue, the nonrenewable magnetization of HXe, and the unique dynamics of the deformable lung during breathing. The final part of this review, Section 8, departs from the conceptual scheme of following intercompartmental transport to discuss these practical considerations, often common to gas and dissolved HXe imaging, and how their consideration can facilitate high quality, quantitative imaging.

## 2. Gas Dynamics during the Breathing Cycle

The functioning lung requires gas from outside the body, with its atmospheric component of 21% oxygen, to be brought in contact with circulating blood from the right heart to support efficient O_2_/CO_2_ exchange. Although subtly different from oxygen because of its greater mass and electric polarizability, xenon gas is a good surrogate for visualizing the flow and diffusive processes by which this exchange takes place.

Gas replacement in the lung occurs via a combination of turbulent flow in the upper airways, transitioning to laminar flow in the lower airways. In the smallest airways and gas exchange units (alveoli), concentration gradients are primarily resolved through gas diffusion. Transitions among these zones depend on inflation level, the rate at which gas is inhaled or exhaled, gas density, subject posture, and disease.

Given the diversity of processes of interest, a wide variety of techniques have been implemented to visualize gas replacement during ventilation. The simplest of these is focused on identifying ‘ventilation defects’: regions in the lung that do not show significant gas replenishment during a typical breath. Defects may result from several different underlying causes, including airway occlusion by mucus or edema, airways closure by bronchospasms, airway collapse in emphysema, abnormal tissue compliance, or other mechanical restrictions resulting in tissue that fails to expand during inhalation

Figure 1 depicts this process in a mouse model subject to the most extreme form of defect, in which an entire lung region is completely unventilated. Here, the mouse model of Invasive Pulmonary Aspergillosis results in localized airway obstruction which is easily visualized by comparing structural images of aerated (gas-filled) tissue to HXe images of tissue in which that gas is efficiently replaced by ventilation. 

In such cases, visual inspection is sufficient to both identify and localize the obstruction. However, in most instances a more nuanced and quantitative assessment of abnormal airflow is desirable. Several refinements have been proposed and implemented, which can be broadly classified into those based on steady-state signal intensity, and those based on signal dynamics between or during individual breaths.

The first category is exemplified in Figure 2: here, a single static image is used to determine if individual regions of tissue are adequately ventilated by comparing the local signal level to that observed under identical conditions in a healthy subject. This process is complicated by the fact that a healthy subject displays a fairly wide distribution of signal intensities (top row) due to a variety of factors, including partial volume imaging effects, an unusually high or low voxel tissue fraction or the presence of sub-voxel vascular or airway structures. Thus, it is generally impossible to determine if an individual voxel’s abnormal intensity is due to disease, artifact, or is simply in the tail of the healthy distribution.

Nonetheless, it has been shown in preclinical [20] and clinical studies [21,22,23,24] that the distribution of intensities is substantially altered in disease. In the exemplified case (Figure 2, bottom row), low-intensity regions in the peripheral lung give rise to a bimodal distribution (left) characteristic of, in this case, the rat model of pulmonary hypertension. In order to visualize the location of these partial defects, the ventilation maps (right) utilize a six-color scheme. Voxels that are determined to be within the lung boundaries and outside of major airways (based on comparison to ^1^H images) are classified as defect (red, intensity I < μ − 2σ of healthy distribution), partial defect (orange, μ − 2σ < I < μ − σ), normal (dark green, μ − σ < I < μ and light green, μ < I < μ + σ), hyperventilated (light blue, μ + σ < I < μ + 2σ and dark blue, I > μ + 2σ). Note that some of the dark blue voxels instead represent small airways that were not removed during airway segmentation, rather than an abnormal or pathological condition.

As with the simple, binary assessment of ventilation defect in Figure 1, this approach allows quantification of the volume of completely unventilated regions. In addition, however, it offers sensitivity to underventilated and overventilated lung regions that may be characteristic of earlier or less severe disease. Furthermore, the scheme is simple to implement and is easily translatable to clinical use, supporting direct comparison of both the extent and spatial patterns of altered airflow. 

On the other hand, single-image approaches are inherently unable to distinguish between diminished signal due to low residual gas volume and that due to inefficient gas exchange during breathing. For that reason, a second category of techniques has been implemented that involve comparison of multiple images acquired at different times during an extended series of breaths. All of these methods use a model of local gas replacement during ventilation, depicted schematically in Figure 3. Here, we model signal dynamics in a region of lung parenchyma (e.g., an imaging voxel) which for simplicity is taken to have unit volume at end exhale. A volume of gas TV with signal per unit volume S_0_ flows into the region and is added to functional residual capacity FRC, which is defined as the gas volume at end-exhale. The fresh and residual gas volumes are assumed to mix, and volume TV of the mixture is then exhaled.

In this context, the signal evolves during inhalation as the local volume V increases from 1 to 1 + TV:(1)$$\frac{dS}{dt}=\frac{dV}{dt}\left(S_{0}-\frac{S}{V}\right)-\Gamma S$$
and during exhalation, as *V* decreases back to 1, as:(2)$$\frac{dS}{dt}=\frac{dV}{dt}\left(\frac{S}{V-1+FRC}+\frac{S}{V}\right)-\Gamma S$$
in which Γ refers to the instantaneous rate of signal loss, primarily due to RF excitation during imaging and collisions between HXe and oxygen gas [25]. The *S*/*V* terms in each expression account for the dilution/concentration of contrast agent as the lung expands or contracts relative to a fixed imaging region.

The above expressions may be numerically integrated and compared to the observed signal dynamics to determine the FRC and volumetric expansion of each region throughout the breathing cycle, as shown in Figure 4. Conceptually, it is straightforward to see how measurement of signal dynamics allows us to disambiguate the cause of locally reduced signal intensity; poor per-breath gas replacement will be characterized by a slow signal buildup over multiple breaths (or equivalently, a small difference between end-inhale and end-exhale signals) because a small fraction of the residual gas is replaced during each breath. Conversely, a low signal region with normal dynamics must be caused by a small regional gas volume.

To focus on the former case, which has a clearer relationship to pathology, a unitless measure of local ventilatory gas exchange efficiency was introduced [26] and investigated in a variety of disease models [27,28,29]. Most of this work utilizes a simplified scheme in which all imaging takes place at either end-exhale or end-inhale breath hold, for which a closed-form solution to Equations (1) and (2) can be found [30]; images are acquired after each breath as the imaging gas is introduced or washed out (Figure 5), and the relevant expression is then solved for ‘specific’ or ‘fractional’ ventilation (SV = TV/FRC, FV = TV/(FRC + TV)).

These metrics have been shown to be significantly more homogeneous in the healthy lung than is observed for the raw signal intensity maps of Figure 2, although gravitational gradients are seen anterior-to-posterior (Figure 6) [31], particularly in larger species, as predicted due to dependent lung compression and perfusion gradients [32,33].

Figure 7 provides an example of the subtle changes in ventilation observable using this technique, comparing the fractional ventilation maps observed in a healthy, free-breathing mouse (A) and an age- and strain-matched mouse with tumor in the posterior, lower right lobe (B). As expected, and similar to the qualitative defects of Figure 1, there is no ventilation in the solid tumor region; this is depicted in the FV histogram as the leftmost yellow bar. Additionally, however, patterns of ventilation throughout the rest of the lung are altered leading to a bimodal distribution. Regions anterior to the tumor are noticeably underventilated, constituting a narrow distribution with mean approximately 15% less than that of the healthy mouse. This is likely due to tissue compression in otherwise healthy, proximal lung. On the other hand, the rest of the organ is overventilated, leading to a distribution with mean ~15% above that of the healthy mouse. This change may represent a physiological response needed to maintain sufficient O_2_/CO_2_ gas exchange in the partially compromised lung.

## 3. Diffusive Dynamics in the Lung Parenchyma

NMR and MRI offer a variety of opportunities to impose diagnostically informative contrast, beyond the imaging of position and flow of the agent explored in the last section. The first of these to be explored in detail is the ‘Apparent Diffusion Coefficient’ (ADC) in an effort to provide sensitivity to parenchymal microstructure below the size limit of any current imaging modality.

There is good reason to suppose that the ADC contains information relevant to lung microstructure and function. Diffusive resolution of airway-alveolar concentration gradients is known to be largely responsible for the final step of gas transport within the acinar structure [34], allowing fresh gas to contact the gas exchange membrane; a disruption of this process is likely to affect the efficiency of that exchange. Additionally, parenchymal changes that lead to diffuse hyperintense ‘ground-glass’ or hypointense regions cannot be reliably diagnosed precisely because the underlying changes are not spatially resolved. Observing how gas diffuses through such structures provides another avenue to distinguish intact, hyperinflated tissue from emphysematous remodeling, or among the different forms of consolidated tissue.

An ADC measurement can be incorporated into any MR imaging or spectroscopy sequence. It requires only the application of a bipolar gradient pulse between the RF excitation and readout, as depicted in Figure 8. The effect of this addition is to dephase, and then rephrase spins that are motionless during the pulse. For spins that undergo diffusive motion, however, the dephasing and rephasing pulses do not perfectly cancel, and a net loss of transverse magnetization (and therefore signal) is observed. In lung tissue, the majority of gas is contained within alveolar spaces with characteristic dimensions of ~30–200 μm, depending on species [35,36]. For comparison, atoms diffusing in a uniform gas sample move through an average distance $\sqrt{2Dt}$ along the direction of the applied gradient in the time *t* between the gradient pulses, subject to the gas diffusion coefficient D = 0.07–0.18 cm^2^/s [37] (depending on the fraction of inhaled xenon). Thus, for *t* > ~100 μs, the dephasing effect of the applied pulses is diminished due to restricted motion in collisions with the alveolar septa. 

This effect is generally quantified by calculating the expected signal reduction for a given gradient pulse configuration, and reporting the (reduced) diffusion coefficient that would have resulted in the observed reduction if the gas were subject to free diffusion. This is the Apparent Diffusion Coefficient. In the limit of short gradient pulses of length separated by time *t*, it has been shown that the expected fractional signal reduction due to the pulses is:(3)$$\frac{S}{S_{0}}=exp\left(-{\left(\gamma G\delta \right)}^{2}t\times ADC\right)$$
in which γ refers to the gyromagnetic ratio of ^129^Xe and *G* to the applied gradient (a more complete treatment, incorporating realistic gradient pulse shapes can be found in ref. [38]).

Healthy lungs, regardless of species, are characterized by an ADC representative of the inhaled gas mixture’s free diffusion coefficient in the large airways, and a largely uniform but greatly reduced ADC in the parenchmyal structure as depicted in the superior regions of Figure 9. In disease, and particularly in emphysema, this uniformity is lost. Alveolar airspaces are distended due to destruction of the septal walls, in extreme cases resulting in emphysematous bullae, which lack alveolar structure and in which diffusion is entirely unrestricted, resembling that of the airways.

Several authors have studied these effects in model animals [41,42]. This work is exemplified in Figure 10, in which the digestive enzyme elastase is instilled into a rat lung, causing emphysema-like tissue degradation. Note that as with many instances of human disease, the enzyme’s effects are inhomogeneous (likely due to trapped air and nonuniform delivery during instillation). As predicted, all measured values are significantly less than would be expected for unrestricted diffusion, indicating that collisions with the alveolar structure substantially limit diffusion, even in the case of widespread damage in the emphysema model. However, the overall effect of the model is an increase in both mean ADC and the standard deviation of ADC values. It is important to note, however, that a similar effect can be observed in the healthy lung under conditions of atelectasis, or localized alveolar collapse. This is of particular concern in anesthetized and ventilated preclinical models, in which the spontaneous sigh reflex is suppressed. These dynamics are exemplified in Figure 11, in which an anesthetized rat underwent periodic ADC imaging interspersed with alveolar recruitment maneuvers intended to mimic spontaneous behaviors in awake animals and humans.

Counterintuitively, pressurization of the airways throughout the breathing cycle causes the mean ADC to decrease, despite the overall increase in lung volume which would be presumed to be accompanied by an increase in size of each alveolar sub-unit. This can be understood only if localized alveolar collapse causes an overdistension of nearby structures that exceeds that associated with the elevated pressure during the recruitment maneuver itself. This result has important clinical implications [43], justifying the application of Positive End-Expiratory Pressure to ICU patients to prevent injury propagation, as is commonly done [44]. In the context of preclinical measurements, it also highlights the need to account for and minimize atelectatic changes to achieve reproducible experimental results.

When interpreting ADC images quantitatively, it is also important to note that the simplified treatment discussed above does not satisfactorily account for all of the ways in which diffusion in the tortuous parenchymal structure of a lung differs from free diffusion in a uniform environment. As might be expected, it has been shown that there is a slight anisotropy in ADC based on the relationship between the predominant local airway direction and the direction of the diffusion-encoding gradients [45,46].

More importantly, the ADC value derived from Equation (3) will differ as the gradient pulse shape and spacing is changed, even if the prefactor multiplying the ADC is unaltered. This phenomenon occurs because diffusion distances in the lung parenchyma do not scale with time in the same way as they do in a homogeneous gas. Although these specialized dynamics may lead to some confusion when comparing ADC measurements in the literature, it can also be utilized to elucidate different aspects of the branching microstructure in the parenchyma.

This approach was developed in a series of theoretical and experimental papers [38,46,47,48]. It is based on the idea that in an inherently anisotropic alveolar structure, in which diffusion along a terminal airway becomes restricted at longer time-scales than that across the airway or the alveolus, multiple relevant structural features can be quantified separately by probing diffusion at a range of times. The set of measurements is then compared to analytical or simulated results derived from a simplified model of alveolar and small airway geometry.

Measurements of this type have a clear advantage in that they are more easily compared between sites; if the same base model is used, and if each set of measurements spans the range of diffusion times needed to constrain it, then the derived morphologic parameters should be largely independent of the details of the individual pulse sequences. Broadly, two approaches have been pursued: the use of at least two measurements to yield two representative ADC values, representing diffusion along (DL) and across (DT) the terminal airway, and the use of multiple (often six) measurements and comparison to modeling to yield estimates of alveolar diameter (L), depth of protrusion from the airway (h), terminal airway radius (r), and/or the mean linear intercept (LM) for comparison to quantified histological sections.

These two approaches are exemplified in Figure 12 and Figure 13, respectively. Figure 12 displays a comparison between ADC measurements in a healthy rat and one subject to lung inflammation and fibrosis in a radiation therapy/radiation-induced lung injury model. The reduced ADC is characteristic of models in which airway lumen is reduced or alveolar septa are thickened due to accumulation of infiltrates or fibrotic remodeling, thus reducing the characteristic dimensions through which gas can diffuse in the parenchyma. Notably, the transverse diffusion coefficient appears to be more sensitive to structural remodeling, which has generally been found to be the case [46,49].

In contrast, Figure 13 displays the increased ADC observed in a rat model subject to elevated inspired oxygen concentration (FiO2). In this case, exposure to 85% oxygen for 48 h caused Hyperoxic Acute Lung Injury [51] with associated alveolar collapse and emphysema-like changes. Both of these features of the acute injury can be expected to lead to an overall increase in gas diffusion, as was observed, but the morphological analysis pursued in ref. [50] allows a more nuanced interpretation of the relative contribution of local distension proximal to atelectasis or edema, which is likely reversible, and septal wall destruction, which is not.

## 4. Spectra and Compartments

The previous two sections outlined how the imaging of xenon and its dynamics during the breathing cycle can elucidate gas motion in response to pressure and concentration gradients through the breathing cycle, in analogy with the flow and diffusion of oxygen and CO_2_. Unlike the other commonly-used hyperpolarized gas, ^3^He, xenon shares another feature with those respiratory gases in that it readily dissolves in tissue and blood. Further, HXe is distinguishable in a variety of tissues based on individual, characteristic resonance frequencies. Thus, similar methods can be employed to discern both the location of dissolved xenon and its transition among these physiologically relevant compartments.

Within a few milliseconds of gas arrival in the alveolar space, distinct spectral peaks appear approximately 200 ppm downfield from the resonance frequency of ^129^Xe gas. An example NMR spectrum in a healthy rat, acquired shortly after HP xenon inhalation, is shown in Figure 14. Note that in addition to the gas in the lung airspaces, which is typically used as a frequency reference and set to 0 ppm, two additional peaks at ~197 and ~210.5 ppm grow over a period of several hundred ms. From in vitro and blood sample studies, these peaks have been found to correspond to ^129^Xe dissolved in pulmonary tissue/plasma and bound to hemoglobin inside the red blood cell, respectively. In the literature these distinct compartments are referred to as the ‘T/P’, ‘membrane’ or ‘barrier’, and ‘red blood cell’ or ‘RBC’ peaks, respectively.

The large chemical shift associated with xenon dissolution has been studied for decades, and the scale observed in vivo is consistent with in vitro measurements in aqueous solutions, lipids, biological extracts, hydrocarbons and polymers [52,53,54]. Although unlike oxygen and carbon monoxide, xenon does not bond directly to the iron atom, the RBC peak has a comparatively large chemical shift, which is thought to originate in the proximity of some xenon binding sites to these centers in hemoglobin.

Interestingly, the individual dissolved resonance frequencies differ somewhat among animal species, to the extent that two distinct peaks are not always discernible. The RBC peak also shifts with local oxygen concentrations due to conformational changes during oxygen loading and unloading, as seen in Figure 15. Although some work has gone into understanding these interactions, and into the location and dynamics of xenon-protein interactions in general [55,56] the reason for in particular the species dependence is not well understood. Nonetheless, some of the most interesting and clinically relevant measurements depend on a separate RBC peak, which is not observed at all in mice [55], and is likely too small to be useful in rabbits and pigs.

Because of the different binding locations, it is tempting to assume that xenon acts as a passive reporter of the presence of accessible hemoglobin. However, Lepeshkevich, et al. [54] have pointed out that in some cases oxygen must pass through the same hydrophobic pockets that bind xenon to reach the iron center, and that the presence of xenon can therefore slow oxygen loading and in particular unloading. The practical importance of this effect is not well understood.

Within approximately 100 ms after inhalation, dissolved xenon accounts for ~2% of the total HXe signal [59,60], although this fraction can be significantly lower with the reduced tissue density of emphysematous disease [61], or higher in the denser lungs of fibrotic disease or small animals [62]. The limited amount of dissolved xenon, along with the short T2* characteristic of lung tissue [63], makes direct imaging of either dissolved compartment challenging. However, it is important to note that inter-compartment exchange leads to rapid replacement of the dissolved signal, allowing higher flip-angle interrogation than would otherwise be the case and correspondingly higher signal-to-noise than the dissolved concentrations would suggest, as per Section 8.

A cursory examination of the dissolved peaks shows that at least in species for which they are distinguishable, they are of comparable width and amplitude, although the RBC peak is generally smaller and sometimes slightly broader. Further, signal destruction caused by excitation of the dissolved peaks is communicated back to the gas compartment and can be used to indirectly probe compartment size and exchange as described more fully in Section 6.

The first step in showing diagnostic utility is identifying features that change in disease models. An example application is shown in Figure 16, in which a rat monocrotaline model of pulmonary hypertension displays a significantly reduced RBC to membrane signal ratio. Similar results have been seen in a rat bleomycin model of pulmonary fibrosis [64] and a model of tissue damage during radiotherapy [65].

It is, however, desirable to interpret the spectra in a more nuanced and physiologically relevant way. This depends on measuring the timing of the uptake process. Fortunately, the rate at which NMR spectra can be acquired is consistent with gas diffusion and blood flow dynamics in the lung, as exemplified in Figure 17 (note that as in Figure 14, these spectra are selectively excited, greatly over-representing the dissolved fraction for better visualization).

The goal, then, is to construct a physical model of lung gas exchange, parameterized using structural or physiological values expected to change in disease. An example of this process appears in Figure 18, in which the gas-exchanging parenchymal structure of a rat is represented as a 1-dimensional slab of capillary blood surrounded on both sides by a wall of tissue. By adjusting model parameters, in this case the physical dimensions of the structural components, it is possible to express dissolved HXe dynamics in terms of microstructural features that cannot be otherwise imaged, and which post mortem histology has shown to be restructured in disease. Further discussion of this process appears in Section 5.

Exploring the spectral dynamics further, we find that multiple additional peaks appear as xenon is carried to and dissolves in distal tissues. As expected, all perfused tissues display at least the two peaks corresponding to membrane and RBC, but in general at least three additional peaks also appear after sufficient time has elapsed that the perfused tissue has taken up enough xenon to be visible. As in human, rat brain displays a total of five peaks (ref. [67] and Section 7), and even though the mouse membrane and RBC compartments are not resolved, whole-body spectra display at least six [59].

Since they require transport to and dissolution in tissues beyond the cardiopulmonary system, these peaks appear over a much longer timescale, as shown in Figure 19A. Here, whole-body mouse spectra were acquired with a flip angle of 90 degrees and with increasing repetition time, allowing the xenon to travel increasingly far from the pulmonary site of gas uptake. Note that unlike in the rat, short-time dissolved spectra consist of a single broad peak in the dissolved region. This contribution persists over time, but is joined by a narrower peak at approximately the same frequency, representing vascular blood, after <1 s, and four additional peaks specific to other tissue types within a few additional seconds.

There is not a direct correspondence between anatomical designation (e.g., organ) and spectral peak since a functional anatomic unit is in general a combination of local chemical environments, such as lipid-rich, parenchymal, muscular or vascular. However, this situation can be clarified using spatially-selective spectroscopy, as shown in Figure 19B. Three compartments localized to the large vasculature display qualitatively similar spectra, but become significant at progressively later times, representing pulmonary blood’s sequential arrival at the heart, aortic arch and descending aorta, respectively. Other structures receiving dissolved gas from the vascular blood, such as the heart wall, kidneys and lipid deposits, also show distinct spectral features, but only after significantly longer delay times, as expected (Figure 19C).

Although unexplored in disease models, it may be that altered spectral characteristics/dynamics in specific tissues is indicative of pathological perfusion or tissue composition changes. Notably, these sorts of studies are much better suited to the preclinical rodent models than in large animals or humans, because of the favorable relationship between the short circulation times and the ~7 s [68] duration of the hyperpolarized state in blood.

A further exploration of these spectral features and their extension to imaging applications, appears in Section 8.

## 5. Quantification of Gas Dissolution and Uptake

The magnetic properties of diffusive molecules in porous media have, for a long time, been known to be descriptive of microscopic properties such as porosity and surface-to-volume ratio (SVR) [69]. Xenon gas, being chemically inert, is an ideal diffusive molecule to measure these systems’ transport properties. Analytical descriptions of time-dependent hyperpolarized HXe MRS was first developed to describe transfer in ‘hard’ porous media such as rocks or ceramics [70,71] and later extended to allow for significant diffusion into ‘soft’ media such as human and animal tissues [72]. An example of a soft porous media is alveoli in the lungs—the region where gas exchange between inhaled air and pulmonary blood occurs. Inhaled HXe gas likewise diffuses into parenchymal tissue and blood at the alveoli, thereby making it a powerful contrast agent for measuring the gas exchange process.

Time dependent in vivo gas exchange was first measured in the canine chest using a chemical shift saturation (CSSR) sequence by Rupert et al. [73]. In a CSSR sequence, the entirety of the dissolved-phase polarization is depleted by a 90° RF pulse. During a delay period, often designated by τ, the dissolved-phase signal is replenished by exchange with hyperpolarized gas in the lungs. Following the delay period, the MR signal is acquired. By varying τ, a CSSR experiment can therefore measure the rate at which HXe gas dissolves into the parenchymal tissue and blood. CSSR is the predominant sequence used for measuring the rate of HXe gas exchange but, as will be discussed in Chapter 7, another sequence known as XTC can also be used. In addition, a combined diffusion weighted and CSSR sequence, termed DWCSSR, has been demonstrated in a proof-of-concept study involving rats [74]. Typical dissolved-phase signal uptake curves normalized to the gas signal and plotted as a function of τ are shown in Figure 20.

As can be seen, short delay times are characterized by a rapid increase in dissolved-phase signal then, after some time, the curve flattens and the dissolved-phase signal approaches saturation.

### 5.1. Models of Gas Exchange

There have been several 1D models developed that relate the uptake curve to qualitative descriptions of the gas-exchange components (i.e., lung morphology and blood flow). The simplest model was developed by Driehyus et al. [66] and later expanded on by Patz et al. [76,77]. The Patz model, as it is commonly referred to, considers the dissolved-phase signal as a single compartment rather than separate RBC and membrane signals in order to improve the signal-to-noise ratio of the spectra. The geometry considered in this model, shown in Figure 21A, is a dissolved-phase slab of finite width with an initial HXe magnetization of zero between two gas-phase regions and with blood flow perpendicular to the gas-membrane interface. Analytic solutions to this model yield measurements of SVR, septal thickness, and the transit time of blood through the gas exchange region.

Månsson et al. derived an analytic solution for gas-exchange considering 1D diffusion in a symmetric circular ‘alveolus’ (Figure 21B) [78]. Unlike the Patz model, the Månsson model considered the RBC and membrane compartments separately, decreasing the SNR but allowing for a more detailed analytical solution and providing measurements of the total diffusion length, capillary diffusion length, tissue thickness, perfusion rate, mean transit time, relative blood volume, and haematocrit. Notably, however, the Månsson model does not provide an estimation of SVR—A critical morphometric measurement.

The most complex 1D model was developed by Chang et al. and named the Model of Xenon Exchange (MOXE) [79]. MOXE (Figure 21C) adopts a similar geometry to that of the Patz model but considers the RBC and membrane compartments separately. A notable improvement of MOXE in comparison to the Månsson model is that the MOXE model fits the RBC and membrane signals simultaneously which preserves information and constraints imposed by correlation between the two dissolved-phase compartments. The physiological parameters calculated by MOXE are the capillary transit time, alveolar septal thickness, air-capillary barrier thickness, SVR, and haematocrit.

As shown in Table 1, septal wall thickness and haematocrit are the two most commonly studied measures with these models and, so, it is worth expounding on their formulation and physiological relevance. The alveolar septum is a thin wall that separates adjacent alveoli. The thickness of the alveolar septum has two effects on the CSSR uptake curve. A thicker alveolar septum limits the rate at which HXe gas can dissolve into the blood and thus affects the initial slope of the uptake curve. Secondly, the finite thickness of the septum affects the saturation capacity of the lung tissue which is presented by the latter flat portion of the CSSR curve. Septal thickness can be affected by various lung conditions including interstitial lung disease and radiation induced lung injury.

Haematocrit is the percentage volume of red blood cells in the blood and, thus, affects the oxygen (and xenon) carrying capacity of the blood. Hence, haematocrit is calculated using the ratio of barrier to red blood cell signals (and cannot be calculated from the Patz model that considers both compartments together). Low haematocrit is a sign of anemia and can be caused by cancers such as leukemia or by other conditions such as radiation induced lung injury.

A comparison of the different models by Stewart et al. found that all three models were in good agreement with each other and acquired physiologically relevant parameters [88]. The goodness-of-fit of the Patz and MOXE models outperformed that of the Månsson model but all models were reported to have acceptably strong fits to the data. It was observed that the extra fit parameters of MOXE in comparison to the Patz model could result in unrealistic morphometric parameters that required restrictions during the fitting process. Therefore, it was suggested that the MOXE model could be improved by fixing some values with a priori knowledge (e.g., haematocrit from blood sample draws).

Because the gas-exchange variables estimated by the aforementioned diffusion models are difficult to interpret relative to typical clinical measurements, models have also been developed to relate the wash-in of HXe to the RBC and membrane compartments to the diffusion capacity of carbon monoxide, DLCO [89,90]. While these models provide less granular information with regards to the gas exchange process, they have the advantage of relating HXe gas exchange to a widely used clinical measurement.

### 5.2. Design Considerations

Careful consideration must be given to several aspects of the gas exchange experimental design. Complete saturation of the dissolved signal prior to recovery through gas exchange is necessary for accurate measurements of the curve fitting variables. This is especially important for calculating SVR which depends most on the earliest time points of the uptake curve. Saturation is typically achieved with multiple 90° hard pulses but it has also been demonstrated that sufficient saturation can be achieved with a custom binomial-composite pulse [91].

Consideration must also be given to the inflation level which can have a strong effect on alveolar morphometry [92]. Although it is easier to control in an anaesthetized and ventilated animal than in humans, variability in lung size between healthy and diseased populations could lead to different levels of lung inflation for a given tidal volume. It has also been shown that tissue geometry and its deviation from the idealized 1D models can have a significant impact on the uptake curves [93], implying that variations in lung structure could confound calculations of morphometric parameters such as septal thickness and SVR. This would be especially relevant in disease models that are expected to have large effects on the geometry of the terminal airways (e.g., emphysema). Furthermore, it has been shown that depolarization of gas-phase HXe in regions of high gas exchange as well as active gas transport through pulmonary circulation can introduce biases into gas exchange measurements [86].

Nevertheless, HXe gas exchange has been measured in a variety of experiments in both human and animal lungs and it has been observed that, even in humans where there is increased variability (e.g., lung inflation, procedural compliance, etc.), CSSR-derived gas exchange variables are sufficiently reproducible in comparison to standard clinical tools [94].

### 5.3. Applications of Gas Exchange MR in Animals

An advantage of pre-clinical models is that tissue is more easily extracted for histological validation. However, as described in the previous chapter, the chemical shift of HXe dissolved in mouse blood is not distinct from HXe dissolved in mouse tissue. Therefore, modeling gas exchange in mice is restricted to the Patz model [80] or requires the use of transgenic mice [81] so the majority of pre-clinical gas exchange experiments have been performed in rats.

Gas exchange has been measured in a variety of pre-clinical disease models including emphysema [80], smoke [95] and pollution [85] inhalation, radiation induced lung injury [75,82,83,84], and bronchopulmonary dysplasia [87]. As the gas exchange models require repeated measurements at multiple time points, the acquisition has traditionally been restricted spectroscopy but spatially resolved gas exchange imaging has been demonstrated using a spiral-IDEAL acquisition sequence [75]. A summary of the gas exchange experiments performed in animals is shown in Table 1.

## 6. “Xenon Polarization Transfer Contrast” (XTC)-Based Techniques

### 6.1. Introduction

Xenon Polarization Transfer Contrast (XTC) refers to a set of techniques in which the dissolution of xenon into the pulmonary tissue and blood is encoded through selective destruction/inversion of dissolved magnetization. This loss of signal is then read out as the affected atoms exchange back to the gas phase, allowing a form of gas exchange/dissolution imaging while eliminating the need for directly probing the dissolved compartments at all. In certain circumstances, this assessment of diffusion-driven magnetization transfer among compartments gives equivalent results to other measures of gas dissolution while making more efficient use of limited magnetization or imaging time. Figure 22 shows the general scheme employed in an XTC measurement: at a minimum, it consists of a first acquisition, a ‘saturation’ period (during which dissolved signal destruction or inversion occurs), and a second acquisition. The difference between the two acquisitions not attributable to other factors (e.g., the time between the acquisitions or the RF applied to acquire first image itself) is the XTC effect, which is directly quantified as the fraction F_XTC_ by which the gas phase signal has been attenuated.

There are two significant advantages to this approach. First, total magnetization, and therefore signal, available in the gas phase is on the order of 50× larger than that of the dissolved phase, owing to the limited solubility of the gas. And second, MRI of the gas phase is much more straightforward than direct measurements of the dissolved compartments because of the ~10× longer T2* [66] and the lack of multiple resonances whose phase relationship is changing during the acquisition. There is, however, a significant disadvantage as well; use of the XTC effect requires the comparison of two images separated in time, before and after the ‘saturation’ period during which dissolved magnetization is destroyed and exchange back to the gas phase occurs. This presents two separate challenges: the need for additional reference measurements to correct for signal changes independent of the XTC effect (along with signal-to-noise loss associated with subtraction of multiple measurements), and the potential for artifact associated with idiosyncratic behavior, such as motion or intrapulmonary gas flow, that cannot be exactly duplicated during these multiple measurements.

It must also be noted that XTC measurements differ from direct imaging of the dissolved phase in that the former provides sensitivity only to the fraction of dissolved xenon that is exchanged with the alveolar gas. Thus, downstream components representing vascular blood and subsequent uptake into other anatomical structures are not visible. Contrast associated with chemical shift signatures that indicate blood oxygenation or distal tissue-specific compartments is therefore unavailable.

Furthermore, it is not necessarily straightforward to directly tie the extent of the XTC effect (i.e., signal lost during saturation) to fundamental physiological parameters of interest without introducing a model of alveolar septal geometry, dissolved gas diffusion, binding to hemoglobin and capillary bloodflow, the details of which are not well understood. The results can therefore change based on the specifics of the saturation and imaging schemes employed, leading to difficulty in comparing measurements at different sites and under different conditions. Further, despite the potential to selectively saturate the separate dissolved phase compartments [96] and derive fundamentally useful dynamical parameters, there has been essentially no effort toward that goal or comparison to the more direct dissolved phase assessment techniques until very recently.

Nonetheless, certain features of preclinical models have the potential to overcome some of the perceived difficulties. In particular, the ability to enforce a near-perfect breath-hold in the anesthetized animal ameliorates problems associated with motion between images (with the notable exception of cardiac motion), and the ability to signal-average over multiple images reduces sensitivity to idiosyncratic flow or artifacts when comparing single measurements. A subset of these multiple measurements can also be used to probe the XTC effect at different saturation frequencies, powers and pulse timings, thereby constraining the models of gas diffusion and exchange over a larger parameter space than would be practical in human imaging.

Nearly all published preclinical XTC work to date has made use of nonselective saturation of both dissolved peaks (if present) together. This includes the study (in dogs) in which the technique was first introduced [96], and the follow-up studies (additionally utilizing rabbits [97,98] and a wart hog [97]). A thorough theoretical foundation of this work appears in the latter publication, and clinical extensions to more nuanced investigations with multiple saturation schemes appear in subsequent papers and reviews [98,99,100,101,102].

Here, however, we focus on considerations related to preclinical imaging and in particular separate evaluation of the dissolved compartments which, with the exception of a recent paper [103] has not been previously addressed. The XTC effect in this context and its dependence on saturation scheme is best understood using whole-lung, spectroscopic measurements as shown in Figure 23, which can then be generalized to imaging measurements as described in Figure 24. These ‘z-spectra’ show the whole-lung XTC effect observed when applying a range of different saturation frequencies, in this case in a healthy rat. Unsurprisingly, when displayed in this way, gas phase depolarization as a function of frequency bears a striking resemblance to ordinary dissolved phase spectra, showing the same two-peak structure, centered at the same frequencies, and with approximately the same peak shapes and widths, as observed in Figure 14.

However, this correspondence breaks down as the saturation power is increased for two distinct reasons. First, both dissolved peaks broaden as the expected on-resonance depolarization during a typical dissolved compartment residence time approaches unity. This is because the effect of excitation at different frequencies becomes indistinguishable if the power is such that all frequencies cause complete or near-complete depolarization of the entire population of xenon atoms entering a given compartment. Second, the peak corresponding to the hemoglobin-bound xenon compartment preferentially disappears at high power. This effect is observed because entry from the gas compartment into the two dissolved compartments is not symmetrical– dissolved xenon must first enter the tissue-plasma compartment before binding to hemoglobin is possible. When the saturation power is such that significant off-resonance relaxation occurs in the former compartment, even the RBC peak center frequency, then the hemoglobin binding event becomes irrelevant to the overall signal dynamics, and the corresponding peak disappears.

Both effects are clearly discernible in the progressively more intense RF saturation along the nine panels of Figure 23 (note that power is expressed in terms of the applied flip angle per ms to eliminate dependence on a coil-specific calibration, but power is proportional to the square of that RF amplitude). As the applied flip angle rate is increased, depolarization rate and the peak widths both increase, indicating that the expected XTC effect in exchanging regions of the lung is getting stronger, but less selective to the desired compartment. At saturation RF strengths above 30–40 deg/ms, the two compartment become indistinguishable, with the tissue/plasma compartment predominating and making it impossible to provide sensitivity to the exchange between the gas phase and the hemoglobin-bound phase, which is likely the most relevant parameter in determining overall lung function.

Ultimately, this sets fundamental limits on the conditions under which XTC imaging can be useful and quantifiable. On the one hand, the applied saturation RF must be intense enough to cause significant relaxation during the period that it is applied because otherwise its effect will be difficult to disentangle from other relaxation processes or noise. On the other hand, applied power is limited to conditions under which the dissolved peaks are clearly discernible, and the length of the saturation period must be limited to that which is compatible with breath-hold durations that do not perturb physiology. These may be quite short in small animals.

As a practical matter, saturation schemes should keep applied RF intensity below approximately 10 deg/ms to maintain compartment selectivity, and therefore require durations of on order 1 s or longer. The condition on saturation power may be relaxed somewhat by making use of additional measurements that allow estimation of the off-resonance effects, e.g., in the full z-spectra and fits of Figure 23. Such a complete measurement scheme is impractical in an imaging context, but at a minimum two additional measurements can be used: one at the RBC peak frequency plus the peak separation, which stands in for the off-resonance effect of RBC saturation on the tissue-plasma compartment, and one at the tissue-plasma frequency minus the peak separation, which approximates the off-resonance effect of tissue-plasma saturation on the RBC compartment. An example of this scheme appears in Figure 24. Here, the ratio of RBC to tissue-plasma XTC depolarization across the lung of a healthy rat is compared to the ratio observed by direct detection of the two dissolved peaks using Chemical Shift Imaging (CSI).

Although the two kinds of imaging are not exactly the same, and in particular the CSI may contain some downstream, non-exchanging blood components that are not present in the XTC images, we would generally expect both types of image to scale with the size of the respective dissolved compartments. When off-resonance correction is applied, as described above, and a moderate saturation scheme is employed, that is in fact what is observed, lending confidence to the idea that XTC imaging can be used to probe gas dissolution and binding to hemoglobin in a time-efficient and quantitative manner. These results also highlight the care with which the saturation scheme must be chosen; saturation that is too weak leads to inaccuracies related to the small XTC contrast compared to available signal-to-noise, while saturation that is too strong causes inaccurate correction for off-resonance saturation of the RBC peak as mentioned previously.

### 6.2. Saturation Schemes

In this intermediate saturation RF intensity regime, there is still significant latitude to choose a saturation scheme that maximizes depolarization rate while maintaining selectivity between the dissolved compartments. At a minimum, RF applied at the center of one of the dissolved spectral features must have a much reduced intensity at the other. As a general rule, for pulsed saturation with a Gaussian lineshape of intensity
(4)$$I\left(t\right)=I_{0}\exp\left(-\frac{t^{2}}{t_{0}^{2}}\right)$$
and desired selectivity *R*, which represents the ratio of applied flip angle at the targeted compartment to that experienced in the other dissolved compartment, pulse linewidth *t*_0_ must exceed
(5)$$t_{0}>\sqrt{\ln\left(R\right)}/\left(\pi \;\delta f\right)$$
where δf is the frequency separation between the two peaks.

Otherwise, although this has not been experimentally studied in detail, we expect based on general principles that the XTC effect of a series of pulses is largely independent of the pulse shape or spacing, but instead reflects only the average power and the duration of the applied RF. This is because the pulse durations required to maintain the desired compartment selectivity are generally significantly longer than T2*; under these conditions, relaxation per unit time due to the saturation pulse is proportional to the square of the applied RF intensity, or the applied RF power. The above expectation is supported by a Bloch simulation of the effect of saturation pulses of varying lengths $t_{0}$, shown in Figure 25. Here, trains of pulses of varying lengths (x axis) are applied to a model system in which a single dissolved compartment exchanges with the gas phase with a time constant of 70 ms, typical of experimental CSSR measurements (Section 7 and ref. [81]).

With the exception of very short pulses, which are undesirable because of their broad spectral width, we see that the expected relaxation in the gas phase is largely independent of saturation pulse length if the pulse amplitude is adjusted such that the average applied power is kept constant (solid traces). The relaxation does depend on compartment T2*, a feature of the lung structure and the imaging field, but this expected insensitivity to the details of the saturation pulse train is likely to hold under all conditions of experimental relevance, including CW saturation schemes. In contrast, differing saturation schemes which amount to the same total flip angle (dotted traces), and which would be expected to display similar relaxation characteristics under conditions of long T2*, show highly variable results.

### 6.3. Interpretation of the XTC Effect

As described in the previous section, it is generally possible to select an XTC saturation scheme that allows for quantitative evaluation of exchange between the gas phase and, if spectrally resolved, the two dissolved compartments. Once other relaxation processes have been accounted for, the directly measured quantities consist of the two additional relaxation rates resulting from saturation of the individual dissolved compartments at a particular average power level. These quantities are described in the literature in terms of F_XTC_, the fractional depolarization after application of a single discrete pulses (ref. [104] and references therein) and are not directly of interest, but each can be recast as a structural or physiological parameter with clinical relevance.

In the case of tissue-plasma saturation, the most useful interpretation depends on the average saturation power employed (or equivalently, the spacing between pulses of a fixed flip angle). In the case of high power saturation, i.e., near-complete relaxation is expected while the diffusing xenon is still close to the gas/tissue interface, previous authors have derived an expression relating F_XTC_ to the local alveolar surface-to-volume ratio [77]. Note that although these conditions do not allow for compartment selectivity, or sensitivity to the RBC compartment at all (as per Figure 23), such sensitivity is not relevant in this one instance and the expression remains valid for preclinical models in which the spectral features are separate.

In the low power limit as discussed in Section 6.1, for which diffusion throughout the tissue/plasma compartment occurs faster than the saturation-induced relaxation, another expression has been proposed and tested, yielding the local tissue-to-gas volume ratio, based on multiple measurements of F_XTC_ from discrete pulses with different spacings (ref. [97], Equation (3)). For the case of interest here, in which the total saturation time greatly exceeds the time required for dissolved gasses to equilibrate with the alveolar space, the tissue-to-volume ratio can be found more directly by scaling the saturation-imposed relaxation rate $\Gamma_{t}$ during pulse length τ in the tissue compartment by the fraction of time each xenon spends in that compartment:(6)$$F_{XTC}=1-\exp\left(-\lambda_{t}\frac{V_{t}}{V_{a}}\;\Gamma_{t}\tau \right)$$

Note that because of the short T2* in dissolved compartments, the relaxation rate in tissue cannot be directly calculated by considering the saturation pulse shape/amplitude, but must instead by estimated using a Bloch simulation or similar approach.

When interrogating the RBC compartment, the situation is somewhat different, as there is no opportunity for selective, high-power saturation. Thus, it would seem that a metric quantifying the rate at which xenon enters in to the RBC compartment is unavailable, unless an accurate method to account for off-resonance relaxation effects in the tissue/plasma compartment can be found. However, low-power saturation (for which the expected relaxation during a typical RBC compartment residence time is much less than unity) still offers physiologically relevant information about the local blood volume and hematocrit, derived identically to Equation (7):(7)$$F_{XTC}=1-\exp\left(-\lambda_{RBC}\;HCT\frac{V_{blood}}{V_{a}}\Gamma_{RBC}\;\tau \right)$$

Although each of these expressions was developed in a clinical context, and their use in preclinical models has not yet been validated, there is no reason to expect any different behavior as long as the animal model hemoglobin displays distinct spectral peaks.

## 7. Whole Body and Brain Imaging

### 7.1. Whole-Body Gas Uptake and Distribution

Inhaled xenon dissolves through the parenchymal tissue into the blood and is then transported to distal organs via the vasculature. HXe can, therefore, be used as a whole-body MRI contrast agent to image blood flow and dissolved gas exchange to any perfused tissue. Whole body imaging, however, is challenging due to the SNR limitations imposed by xenon concentration dispersion and T_1_ relaxation during transport in vivo, although the latter consideration is less restrictive in small animal preclinical models with short circulation times.

HXe cannot be considered a substitute for structural ^1^H MRI of organs and tissues distal to the lung; this conventional imaging is robustly understood and routinely used in both research and the clinic. However, the magnetic resonance properties of the acquired HXe signal (i.e., chemical shift and relaxation rates) offer qualitatively different opportunities for contrast between tissues, or even functional compartments within the same region. For example, Figure 26 shows HXe spectra acquired in different tissues from the rat body. In the rat brain alone, at least five distinct spectral peaks have been identified (Figure 27), corresponding to HXe dissolved in the red blood cells, plasma, grey matter, white matter, and lipid [67]. Each of these compartments may be present in different concentrations in an imaging voxel, but each must exchange dissolved gas with the blood to sustain metabolic activity. Visualization of these tissue- and compartment-dependent processes (and their alteration in disease) can offer dynamical information that is difficult to get using any other technique.

Figure 28 demonstrates how these properties can be exploited for imaging, depicting chemical shift images in a mouse at different time points as the HXe is transported away from the lungs. In this example, the HXe transit time was controlled by varying the rate of depolarization from RF excitation, thereby imposing an effective T_1_ rate (i.e., T_1,RF_ [57,107]). As can be seen in the figure, differences in chemical shift and transverse relaxation can be used to resolve different tissues and organs in vivo. The venous blood (blue) and the arterial blood (red) are well resolved by differences in their chemical shift. This feature, along with the spectral lineshape, also allows differentiation of HXe dissolved in various tissues from that of the vascular blood that perfuses or drains it. Examination of each tissue type and the rate at which xenon accumulates provides a window into systemic distribution of dissolved gas through perfusion– in this case gas taken up in the lungs appears, in order, in the lung tissue, left heart (within ~200 ms), aorta, vena cava/right heart, kidney/cardiac muscle, liver and lipid (with a characteristic time of ~5 s). It may be that disorders affecting local vascular or perfusion characteristics will induce changes in gas concentration, arrival time or, if oxygenation is affected, chemical shift that are diagnostically informative.

### 7.2. Imaging of HXe in the Rat Brain

One application for which this is clearly the case is the brain. As the brain is highly perfused, imaging with HXe is both technically easier in comparison to other distal organs and is physiologically important. Therefore, other than the lungs, it is the most commonly imaged organ. Brain imaging, in contrast to lung imaging, is unaffected by motion artifacts during breathing. Therefore, imaging sequences can be averaged over many breaths. Furthermore, signal loss due to RF depletion and T_1_ relaxation is compensated for by the wash-in of unexcited HXe in the upstream vasculature. These advantages have made it possible to image the HXe distribution in the rat brain despite the challenges of low SNR.

Stroke, which creates strong regional effects and has robust literature on animal models, is a good candidate to test the ability of HXe imaging in vivo. Zhou et al. used a chemical shift imaging sequence to acquire dissolved-phase images of HXe in the brains of rats following middle cerebral artery occlusion [108]. The images (Figure 29) showed a strong decrease in signal in regions of infarct that correlate well to ^1^H ADC imaging and 2,3,5-triphenyltetrazolium chloride (TTC) staining. The results demonstrated the feasibility of detecting variations in HXe distribution in the rat brain following insults, such as stroke.

### 7.3. HXe Signal Uptake in the Brain

Due to the scarcity of HXe signal in the brain, determining the signal time-course in the brain and its dependence on ventilation parameters and relaxation rates is important for optimizing the signal acquisition. The rate of HXe uptake in the brain is dependent on a number of factors including: the rate of ventilation, T_1_ relaxation and gas transfer in the lungs, the rate of blood flow and T_1_ relaxation in the blood, the rate of gas transfer from the blood into the brain tissue, and T_1_ relaxation in the brain tissue itself. An equation for concentration of HXe in the brain during continuous ventilation was derived by Peled et al. [110]. Later, Kimura et al. and Imai et al. formulated similar equations based on the same principles but accounting for the reduction in hyperpolarization due to RF excitations [111,112]. At steady state, the signal in the brain as formulated by Peled et al., *S*, becomes:(8)$$S=\frac{C_{i}e^{-t_{i}/T_{1\left[blood\right]}}\lambda R_{F}F_{i}}{\left(R_{F}+V_{A}/T_{1\left[lungs\right]}+\dot{\lambda Q}\right)\left(F_{i}/\lambda_{i}+1/T_{1\left[tissue\right]}\right)}$$
where *C_i_* is the inhaled concentration of HXe. *λ* and *λ_i_* are the partition coefficients between gas and blood, and blood and tissue, respectively. *R_F_* is the rate of gas exchange. *F_i_* is the rate of blood flow in the brain. *V_A_* is the alveolar volume and *Q* is the pulmonary blood flow. *T_1[blood]_*, *T_1[lungs]_*, and *T_1[tissue]_* are the longitudinal relaxation times in the blood, lungs, and brain tissue, respectively.

*T_1[lungs]_* decreases with increasing oxygen concentration [113] while *T_1[blood]_* increases with increasing oxygen concentration [114,115] leading to a non-monotonic relationship between *S* and the fraction of inhaled oxygen during ventilation. Li et al. experimentally determined that *S* is maximized at a pulmonary oxygen concentration between 25% and 35% [116].

*T_1[tissue]_* is an important parameter for pulse sequence optimization but its measurement has been confounded by low SNR and physiological effects (e.g., wash-out). Many studies have attempted to measure *T_1[tissue]_* in the rat and mouse brains but the values range from a few seconds to tens of seconds and consensus has not been reached in the field [110,113,116,117,118,119,120]. Likewise, a range of transverse relaxation rates have been measured for HXe in the rat brain. Wilson measured a T_2_ of 4.57–4.72 ms in ex vivo brain homogenate at 9.4 T [114], Duhamel et al. measured a T_2_^*^ of 8 ms for whole brain signal at 2.35 T [121], and Mazzanti et al. measured a T_2_^*^ of 5.42 ms for grey matter signal at 4.7 T [122]. Future improvements in signal strength (e.g., hyperpolarization) and acquisition sequences may improve the accuracy of these measurements which, in turn, could improve image quality and quantification. Nevertheless, as the signal time-course of HXe in the brain depends on a number of physiological parameters such as blood flow and blood-brain barrier partition coefficient, HXe can be used as a contrast agent for functional brain imaging.

Although the Peled model has been the predominant basis for HXe uptake models used in the field, Martin et al. published a similar but different formulation around the same time [123]. While the Peled model began with a ^129^Xe reservoir in the lungs, their model included transport of HXe to the lungs via the mouth and throat and, therefore, can accommodate a variety of breathing maneuvers.

### 7.4. Functional HXe Brain MRI

Function HXe brain MRI was first demonstrated by Mazzanti et al. in an experiment where the distribution of dissolved-phase HXe signal in the rat brain was measured before and after a chemical irritant was injected into the rats’ right forepaws [122]. The results, shown in Figure 30, demonstrated an increase in dissolved-phase signal in the regions of the brain known to be associated with forepaw pain. Furthermore, it was found that the signal difference between baseline and stimulation was comparable to the signal difference measured under the same conditions using conventional BOLD fMRI. The low flip angle required for hyperpolarized chemical shift imaging as well as the relatively low polarization (8–11%) precluded resolving the dissolved signal into its separate spectral components and the long acquisition time precluded imaging of the time course of HXe in the rat brain.

These limitations were recently overcome by using a spiral-IDEAL imaging sequence developed to acquire spectrally resolved time-course images of the rat brain [107]. As shown in Figure 31, spectrally resolved images of HXe dissolved in the brain tissue and in cerebral blood were acquired at various wash-in times. For a given wash-in time, the amount of HXe signal that reaches the brain tissue is dependent on both cerebral blood flow and blood-brain barrier partition coefficient. Therefore, the ratio of the tissue-to-blood signal can be used to measure these physiological properties. To test this sensitivity, the ratio of tissue-to-blood signal intensities was measured during normo- and hypercapnia. (Figure 32) The experiments demonstrated an increase in gas transfer during hypercapnia, consistent with the expected hypercapnic response (i.e., vasodilation).

### 7.5. Hyperpolarized Chemical Exchange Saturation Transfer (HyperCEST) Imaging

A novel modality for dissolved-phase imaging that has only recently been demonstrated in-vivo [124] is hyperpolarized chemical exchange saturation transfer (HyperCEST) imaging. Chemical exchange saturation transfer (CEST) is a molecular imaging technique that selectively excites a solute proton of interest and then measures the transfer of saturation between the excited molecule and the bulk water protons in the body. HyperCEST is an analogous technique that uses HXe signal in lieu of the ^1^H signal. Specifically, the hyperCEST technique involves saturating ^129^Xe atoms caged in supramolecular hosts and then measuring the change in signal as the caged atoms exchange with HXe dissolved-phase atoms outside of the molecule. Although there still exists significant practical challenges while implementing hyperCEST in vivo [125], it has the potential to provide far more sensitive imaging measurements that traditional dissolved-phase HXe imaging.

## 8. Practical Imaging Considerations

Preclinical imaging with HXe differs from both conventional ^1^H MRI and clinical hyperpolarized applications in several important ways that lead to qualitatively different optimal acquisition strategies. Here, we focus on lung imaging, but similar considerations apply to imaging of downstream compartments as well.

### 8.1. Gas Delivery and Animal Handling

Gas delivery for preclinical hyperpolarized gas MRI is among the major challenges of this technique. The gas can be delivered as a bolus via a syringe [41,125] but this procedure limits the imaging to a single breath-hold, and can be challenging in cases where tidal volumes are small (~0.2 mL for mice). Because of physiological restrictions in implementing very long breath-holds in animals, multi-breath imaging is more commonly done. Multi-breath imaging affords the possibility of accumulating signal over several breaths, which allows pushing the limits of resolution and SNR, and also imaging the distribution of xenon in distal organs and tissues which builds up over longer timescales of several seconds. For multi-breath imaging, polarized gas can be delivered via spontaneous inhalation (free-breathing) [57,126,127] or mechanical ventilation [63,107,127,128].

Early research on preclinical hyperpolarized gas MRI investigated several different animal species such as rodents, guinea pigs, dogs, sheep, rabbits, and pigs. Over the past decade, imaging methods and procedures are becoming increasingly more standardized and the experimental focus is moving toward identifying new application areas and clinical translation/reverse translation. To this end, rodents have been preferred owing to the availability of well-established and inexpensive models of disease with high relevance to clinical translation. Therefore, this section will focus on animal handling considerations for rodent imaging. As with most other preclinical MR experiments, hyperpolarized gas MRI is performed under anesthesia and requires monitoring of respiration, temperature and heart rate. In a free-breathing setup, a continuous stream of a gas-mixture composed of polarized gas, oxygen, and gas anesthetic such as isoflurane are delivered via a nose cone, and exhaled gases are scavenged out of the bore (Figure 33). Imaging may be respiratory gated to the full-inspiration or the longer end-exhalation phase. In a mechanical ventilation set up, injectable anesthetics such as pentobarbital, ketamine or xylazine are more commonly used because of the challenges in flowing polarized gas through a vaporizer without substantial depolarization. Mechanical ventilation allows more precise control over positioning the MR trigger for gated imaging over the desired phase of the respiratory cycle.

Commercially available rodent ventilators are generally not capable of delivering hyperpolarized gas owing to unique design requirements such as limiting contact of hyperpolarized gas and either oxygen or materials with ferromagnetic impurities, and being MR conditional. Customized HP gas-compatible ventilators incorporating these design features have been prototyped in house by few groups [63,107,127,128] or through an external commercial arrangement [59], and robust HXe MR images have been demonstrated. With mechanical ventilation, a constant volume scheme is typically employed, and the gas-mixture is delivered via an intubation catheter that is secured with a tracheotomy or inserted orally. For oral intubation, customized catheters may be needed to ensure a gas-tight seal in the trachea. Passively exhaled gases are returned to the ventilator through a low impedance exhale line. Figure 34 shows the main components of an HP gas rodent ventilator that has gas-delivery and physiological monitoring capabilities, and Figure 35 shows its customized intubation catheters, and monitoring display.

### 8.2. Imaging Considerations

Hyperpolarized gas MRI has unique imaging considerations owing to the challenging lung environment where ventilation, gas exchange, T1 relaxation, and susceptibility effects cause substantial and rapid changes in the detected signal. A customized imaging protocol therefore needs to be established to maximize SNR and obtain accurate information about lung microstructure and function under these conditions.

#### 8.2.1. Magnetization Density

First, it is important to understand the practical limits of available xenon magnetization in the lung airspace and tissue compartments. Lung tissue density depends on species and inflation level, but generally varies between 0.2 and 0.4 g/cm^3^. Consisting mostly of water as with most tissues, this gives a proton density of approximately 30 M. To estimate the available proton MRI signal, we multiply this density by the degree of alignment (polarization) generated by the magnetic field and the proton gyromagnetic ratio, as per the first row of Table 2. For comparison, practically achievable HXe magnetization values for the gas and dissolved phase are given in rows 2 and 3. Despite the low density of inhaled gas, hyperpolarization increases gas-phase magnetization to the level of ^1^H MRI. Dissolved phase magnetization is approximately 50× lower than that of the gas phase because of the ~10% tissue density in the lung, the ~10% solubility of xenon gas in tissues [66], and the contribution of hemoglobin-bound xenon [60].

#### 8.2.2. Imaging the Gas-Phase

Inhaled xenon in the pulmonary airspaces is relatively straightforward to image owing to the large available magnetization as noted in the table. Within a single breath, there are 3 detrimental factors that reduce the available magnetization: (1) inhaled xenon gas comes in contact with paramagnetic oxygen, which accelerates its T1 relaxation time from several minutes outside the body to ~19 s in the lungs [112]; (2) Each RF excitation pulse reduces the total available longitudinal magnetization by cosine(α); (3) After RF excitation, the transverse magnetization decays with a short T2* of ~5 ms at 7 T [128].

Several different pulse sequences have been used to image gas-phase xenon such as GRE [42], 3D radial (UTE3D [63,64,129]), spiral [75,130], but all share a few key fundamental implementations to maximize signal. First and foremost, multi-breath imaging mitigates challenges (1) and (2) above by bringing in fresh polarized gas with each breath. One or more lines of k-space are acquired within each breath and so the total acquisition can span over several breaths lasting several minutes. The acquisition is typically triggered to the full-inspiration phase to minimize motion artifacts. The total scan duration is only limited by the volume of polarized gas available and the T1 relaxation time of the hyperpolarized gas in its storage compartment outside the magnet. Typically, the scan time is limited to <20 min to limit substantial differences in the signal from T1 relaxation over the duration of the scan [20,127].

With regard to acquiring data across multiple breaths, the simplest strategy is to implement a single RF excitation within each breath with a flip angle of 90°. More commonly, multiple RF excitations are implemented within a single breath to reduce the total acquisition time, acquire images at higher resolution, or to average to boost SNR. If multiple RF excitations are implemented in each breath, then a different flip angle strategy is warranted to distribute the magnetization evenly between the excitations. A fixed small flip angle can be used for every excitation [48,127,131], which induces a modest amount of depolarization with each excitation. Alternatively, a variable flip angle strategy can be employed [89,132,133,134,135] where the flip angle is gradually incremented from a small values to ultimately a 90° pulse to use up all the remaining magnetization. The latter strategy ensures uniform transverse magnetization across all excitations, but is more difficult to implement.

These considerations are exemplified in Figure 36, which shows the UTE3D pulse sequence, time domain FID data for gas-phase xenon MRI and the resulting image in a mechanically ventilated rat. The short echo times afforded by UTE3D compensate for the short T2* of xenon in the lungs, and the center-out encoding scheme with randomized directions allows undersampling of k-space while maintaining high image fidelity. Imaging was gated to occur over the breath-hold. A fixed flip angle of 15° was used, and 20 radial rays were acquired over each breath-hold. Figure 36c shows the time domain signal within 3 breaths. In each breath, the peak of the FID is observed to gradually decrease with each excitation. With each new breath, fresh magnetization enters the lungs, which resets the signal to its initial high intensity. Figure 36d shows time domain data from the entire acquisition lasting 2.5 min (75 breaths). Over the course of the acquisition the peak intensities of the FIDs are observed to gradually decrease from T1 relaxation of xenon stored in the gas bag connected to the ventilator.

#### 8.2.3. Imaging the Dissolved-Phase

Dissolved-phase xenon imaging requires a fundamentally different set of considerations owing to some key differences in the nature and evolution of magnetization in this compartment: (1) the magnetization in the dissolved compartment is 2 orders of magnitude smaller than the gas phase; (2) the dissolved phase occurs at a chemical shift of ~200 ppm downfield from the gas phase; (3) there is a constant flow of fresh magnetization from the gas phase to the dissolved compartments by virtue of diffusive exchange of gas between the two compartments; and (4) the T2* of dissolved xenon in the lung is very short, ~0.5 ms at 7 T [127].

To address (1) and (2) the transmit/receive frequency must be set to that of the dissolved phase (typically the dominant membrane resonance at ~196 ppm), and a frequency selective pulse must be used to avoid simultaneously exciting the much larger gas-phase compartment, which can introduce substantial chemical shift artifacts. (3) works to the benefit of dissolved-phase xenon imaging because the constant flow of fresh magnetization allows the use of large flip angles even up to 90° to maximize SNR and compensate for its low concentration. The flip angle and the repetition time together control the effective replenishment time of dissolved phase magnetization [108]. At short replenishment times, xenon signal is limited to the gas-exchange region of the lungs. Longer replenishment times allow xenon magnetization to circulate to the heart and further downstream to other organs. Finally, the very short T2* of dissolved xenon is a significant challenge and requires a short TE and rapid readout, which is enabled by UTE3D or spiral pulse sequences with center out encoding coupled with a high receiver bandwidth.

Figure 37 shows gas- and dissolved-phase xenon images in a mouse at 2 T with UTE3D and the parameters in the table. The dissolved-phase image was acquired with a flip angle of 30° and TR of 50 ms. Even though the TR is short, the smaller 30° flip angle increases the effective replenishment time and HXe signal is observed in the left ventricle of the heart and the aorta.

#### 8.2.4. Separating the Dissolved Phase Image into Membrane and RBC Compartments

As described in Chapter 4: Spectra and Compartments, the dissolved phase of xenon at short replenishment times exhibits two distinct resonances corresponding to the membrane and RBC compartments in rats and humans. In rats, these resonances are observed at ~197 ppm and ~210.5 ppm, respectively. It is possible to separate the total dissolved phase image into membrane and RBC components using imaging techniques such as Dixon [134] or IDEAL [135]. These techniques were originally developed to separate fat and water in ^1^H MRI, and over recent years have been widely adopted by the hyperpolarized gas MRI community to separately image the membrane and RBC compartments and through this directly evaluate regional abnormalities in gas-exchange.

The Dixon and IDEAL methods exploit the difference in Larmor precession frequencies between two resonances that results in predictable phase shifts in the transverse plane. A radial or spiral readout is typically employed and involves one to three precisely calculated echo times, at which the two resonances are out of phase by empirically pre-determined margins. The signal from the echoes can then be mathematically resolved into two images corresponding to the two resonances. The simplest implementation is the 1-point Dixon technique, where a single echo time is employed, which corresponds to the two resonances being 90° out of phase with each other (860–940 μs in rats at 2 T [66]; 248 ± 5 μs in rats at 7 T [20]. Then the phase of the receiver can be adjusted such that one resonance corresponds to the in-phase image and the other to the out of phase image [66].

Resolving the dissolved phase image into barrier and RBC compartments provides unique information about abnormalities in gas-exchange that might not be clearly observed in the total dissolved image alone. Figure 38 shows xenon gas, membrane, and RBC images in the rat bleomycin model of pulmonary fibrosis [66]. The RBC image shows total loss of signal in the injured left lung likely indicating that the fibrotic interstitial tissue posed a diffusion limitation for xenon transfer to capillary blood.

As described for ventilation imaging in Section 2, dissolved phased images can also be quantified by binning the signal intensity into different clusters corresponding to hypointense, normal, and hyperintense signal. For example, Figure 39 shows ventilation, membrane, and RBC images in the monocrotaline rat model of pulmonary hypertension (PH) [20]. The PH model showed several defects in the RBC image, as well as some elevated intensities in the membrane image pointing to possibly regions of fibrosis.

## 9. Conclusions

We have presented a review of the capabilities and practical considerations of HXe MRI in the preclinical setting. Although the preclinical set up for this technique is more challenging than in clinical imaging, it provides a vital platform to not only translate/reverse translate clinical research, but to also develop new imaging protocols that more efficiently use the available magnetization and extract increasingly more pathophysiologically relevant information. Moreover, imaging at longer replenishment times can identify new applications in diseases beyond the lung. We hope that the content in this review can serve as a starting point to new sites establishing this capability as well as active sites eager to explore new avenues. As increasingly more imaging sites establish preclinical HXe MRI capabilities, the field will benefit from establishing some standards in animal set-up and gas-delivery, imaging methods, and analysis procedures to ensure the reproducibility and validity of research results.

## Figures and Tables

**Figure 1 molecules-27-08338-f001:**
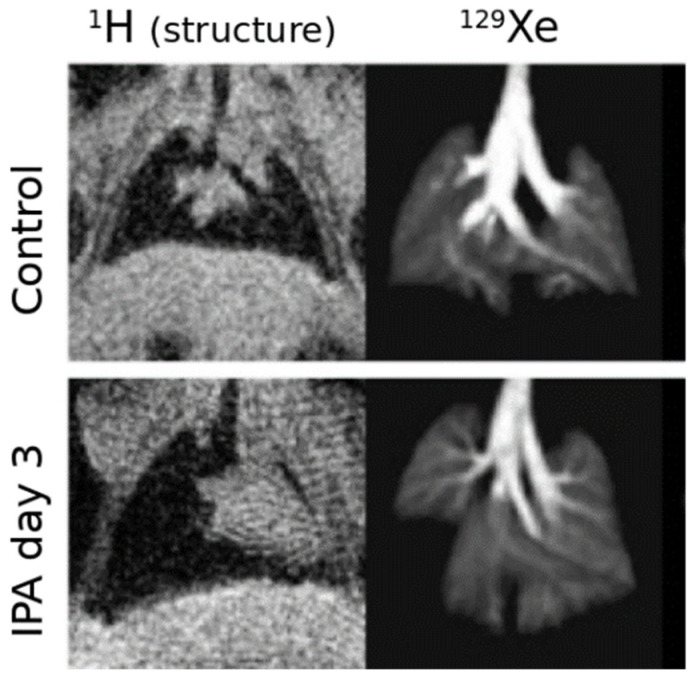
HXe images (right) in a mouse model highlight the obstruction of a major airway due to Invasive Pulmonary Aspergillosis (bottom). As is often the case, structural imaging (left) leaves ambiguity about the extent and location of obstruction, but the lack of signal in the unventilated region shows that the right inferior bronchus is completely occluded.

**Figure 2 molecules-27-08338-f002:**
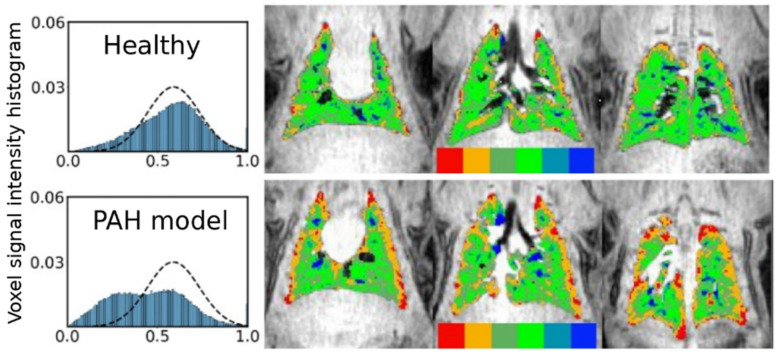
HXe images in a healthy rat (top) and pulmonary arterial hypertension model (bottom) show distinct patterns of reduced ventilation in the disease model (yellow and red partial and full detects). Note that interpretation of reduced signal as hypoventilation requires careful calibration of signal levels between subjects and accounting for partial volume effects during imaging. Reproduced with permission from Virgincar, 2018 (ref. [19]).

**Figure 3 molecules-27-08338-f003:**
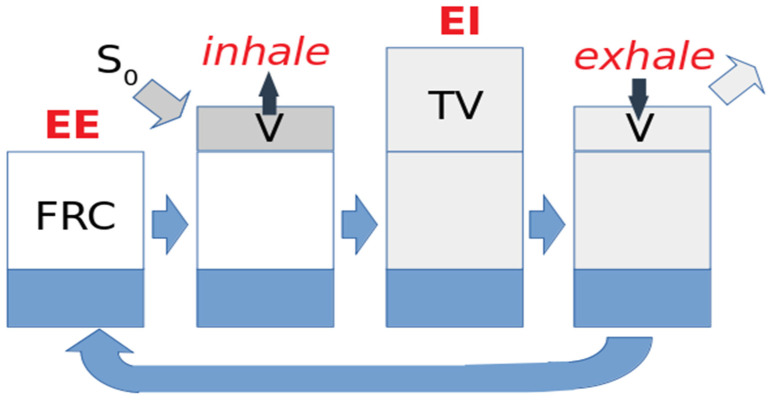
A schematic depiction of gas replacement in a region (voxel) of lung parenchyma. During each breath, a volume of contrast agent gas TV is added to and mixes with the residual gas of volume FRC, and the mixed gas is partially exhaled back to FRC. Note that FRC is less than the voxel volume because with the exception of airways, each voxel contains some tissue. Local gas replacement efficiency can then expressed as the unitless ‘specific’ SV = TV/FRC or ‘fractional’ FV = TV/(TV + FRC) ventilations.

**Figure 4 molecules-27-08338-f004:**
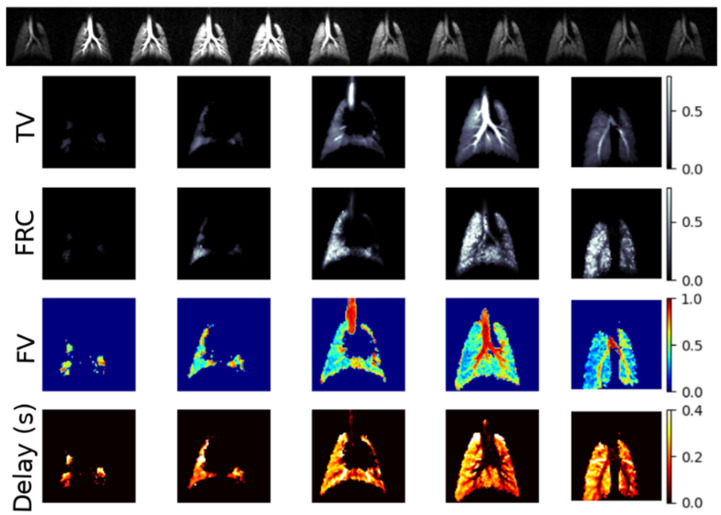
A series of images, assembled from acquisitions during ~100 breaths of 10% HXe during 3 min. free breathing in a healthy pig. The top row shows signal dynamics during a typical breath; signal appears in the trachea and large airways first on inhalation (top row, left) and is distributed to the parenchyma more slowly. Analysis of signal dynamics according to Equations (1) and (2) yields maps of local TV, and FRC (rows 2,3), the ventilation efficiency (row 4), and time required for the parenchymal region to reach full inspiratory volume (row 5). The latter parameter may be particularly useful in highlighting obstructed flow and abnormal breathing dynamics due to compromised lung compliance, musculoskeletal changes or the Pendelluft effect.

**Figure 5 molecules-27-08338-f005:**
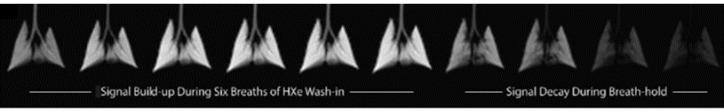
A series of 6 wash-in breaths of 70%/30% HXe/O_2_ is followed by four images acquired over a 12-second breath hold in a horizontal slice in a healthy rabbit. Washin dynamics illustrate the uniformity of signal buildup arising from uniform ventilatory gas exchange efficiency characteristic of health. Images acquired at breath-hold show some shrinkage of the lung as xenon is taken up into the bloodstream, and nonuniformity of decay representing inhomogeneous oxygen concentrations, particularly in the large airways and posterior azygous lobe.

**Figure 6 molecules-27-08338-f006:**
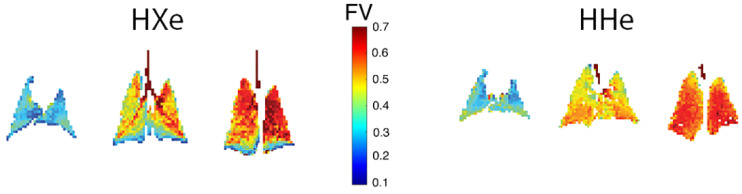
Ventilation inhomogeneity related to posture/gravity and the transition between flow- and diffusion-dominant transport in small airways are known to be present in all species. Gravitationally lower regions experience more efficient gas exchange as in the rabbit model above, and the denser ^129^Xe gas (**left**, HXe) additionally experiences a central-to-peripheral and apical to basal gradient when compared the lighter ^3^He (**right**, HHe). Each set of three images depicts slices from anterior (left) to posterior (right).

**Figure 7 molecules-27-08338-f007:**
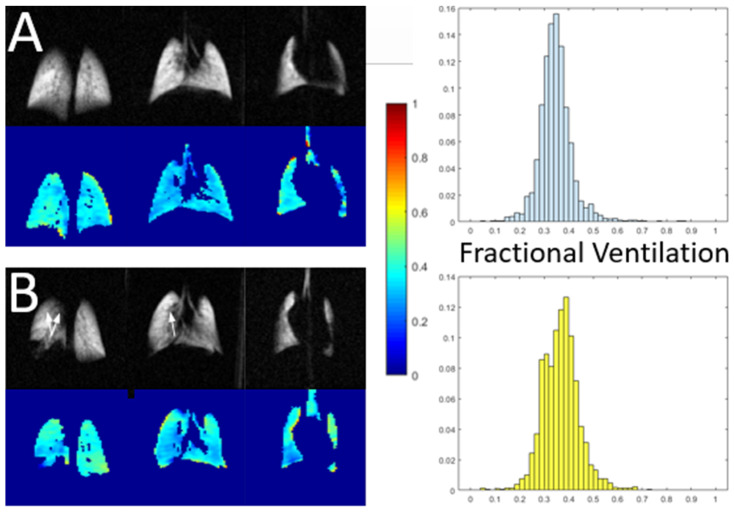
A comparison of local per-breath gas replacement efficiency in healthy (**A**) and tumor-bearing (**B**) lungs highlight the loss of ventilation in the tumor region (last row, lower left) as well as the changes to regional changes due to mechanical effects of the tumor throughout the organ (bottom row and accompanying histogram).

**Figure 8 molecules-27-08338-f008:**
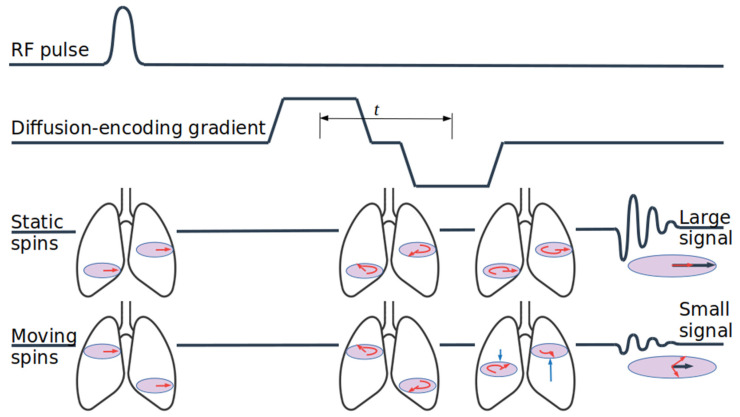
A simplified depiction of diffusion-encoding to measure the Apparent Diffusion Coefficient. An RF pulse (top) is applied to generate a net transverse component of the hyperpolarized spins (bottom two rows, left, red arrows). These spins are then subject to a spatial field gradient, causing the spins to dephase based on position (second row). An inverse gradient is then applied to reverse the spatially-dependent dephasing, resulting in restoration of the full MRI signal (third row). If, however, individual atoms move significantly during or between the gradient pulses (bottom row, blue arrows), the rephasing is not complete, resulting in ca diminished MRI signal from which effective diffusion parameters can be extracted.

**Figure 9 molecules-27-08338-f009:**
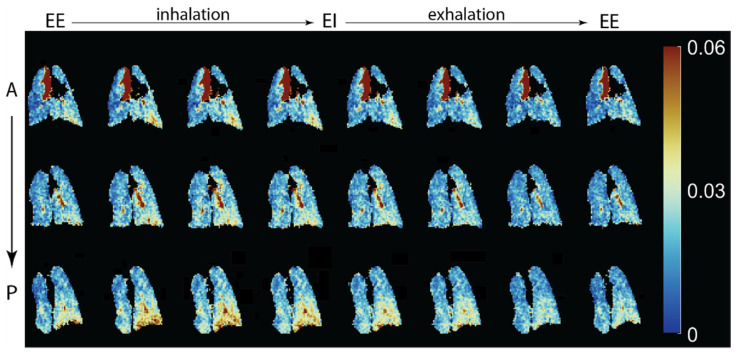
A dynamic map of HXe ADC in a free-breathing porcine model of restriction in thoracic insufficiency syndrome [39] depicts the essential features of this measurement. Free diffusion in the trachea and large airways is reduced by a factor of ~2.5 in the lung parenchyma due to collisions with the alveolar and small airway structure. ADC is modestly increased during inhalation proximal to the diaphragm, but only in the unrestricted lung which is significantly expanded during breathing. The color map depicts local ADC values in the range 0–0.06 cm^2^/s; as noted below numerical values depend on the exact choice of gas mixture, and diffusion-encoding gradient strength and timing, but these qualitative trends are universally observed [40].

**Figure 10 molecules-27-08338-f010:**
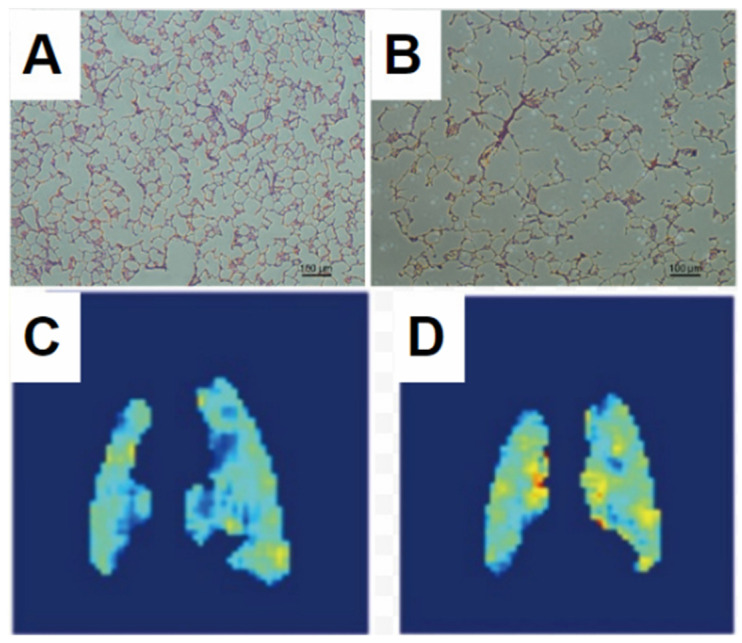
Instillation of the enzyme elastase into rat lungs (**B**) causes focal areas of tissue degradation and enlargement of the relatively uniform alveolar structures characteristic of the healthy lung (**A**). This enlargement can be visualized through ADC imaging ((**C**), control); ((**D**), elastase)) in which both the mean and standard deviation of diffusion distances are significantly increased. Adapted with permission from Wang, et al. (*Magn Reson Med* 2018, ref. [42]), in which the full dataset shows that ADC and histological structural metrics are moderately correlated, and that appropriate thresholds separate the control and model animals using either measurement.

**Figure 11 molecules-27-08338-f011:**
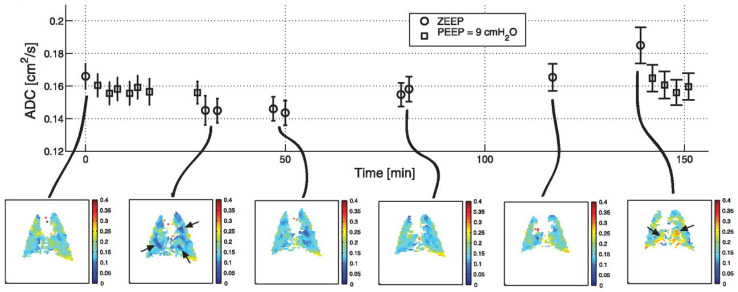
An anesthetized rat, initially imaged after one hour of tidal ventilation, undergoes a ‘recruitment’ maneuver consisting of Positive End-Expiratory Pressure (PEEP), causing the average ADC to decrease. During a subsequent period of 100 min of tidal ventilation, mean ADC increases by approximately 20% due to localized atelectasis. Re-recruitment returns the ADC to baseline. The black arrows highlight regions of regionally reduced (left) or increased (right) ADC that appear immediately before or after recruitment. Reprinted from Cereda, et al. (*J Appl Physiol* 2011, ref. [43]) with permission.

**Figure 12 molecules-27-08338-f012:**
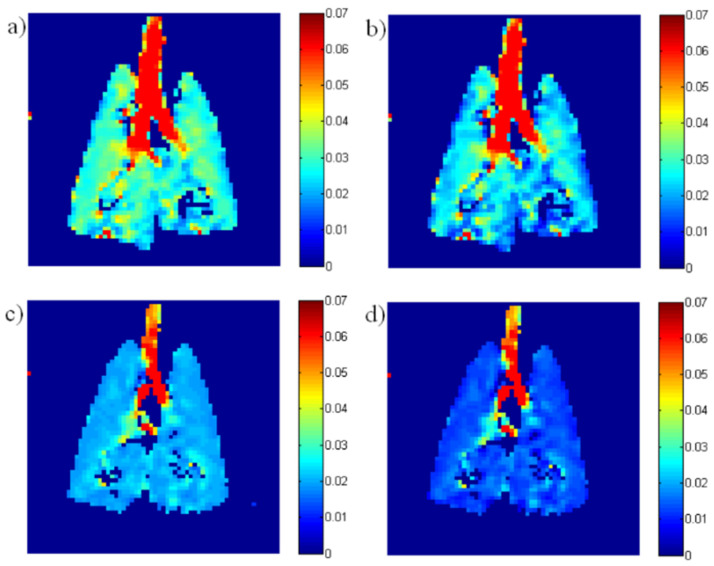
A rat model of radiation-induced lung injury displays a modestly reduced diffusion coefficient along the terminal airway (DL, (**c**)) and a more substantial reduction across the terminal airway/alveolar structure (DT, (**d**)) in comparison to the control animal ((**a**) and (**b**), respectively). Note that both values approach that of free diffusion in the large airways, as expected.

**Figure 13 molecules-27-08338-f013:**
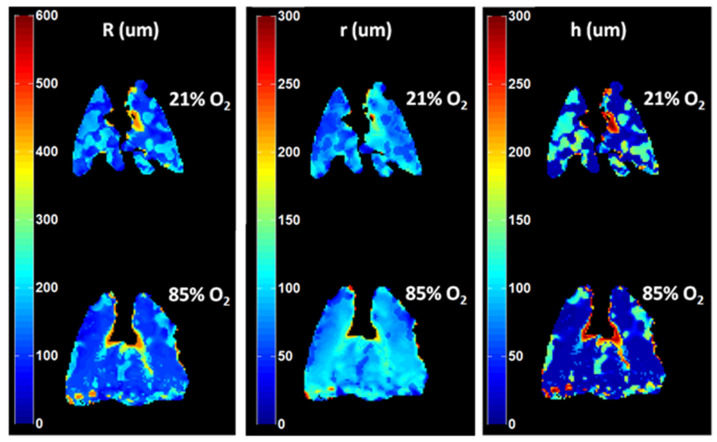
Probing diffusion in the lung parenchyma at multiple time scales allows extraction of several parameters describing microscopic lung structure. In this figure, reprinted with permission from Lindenmaier, 2022 (ref. [50]), alveolar diameter (R), terminal bronchiole radius (r-h) and alveolar depth (h) are separately extracted from six ADC measurements as per ref. [38]. A cohort exposed to hyperoxia (FiO2 = 0.85 for 48 h) experienced edema, inflammation and mild emphysematous remodeling, which was most readily visualized as an expansion of the terminal bronchioles.

**Figure 14 molecules-27-08338-f014:**
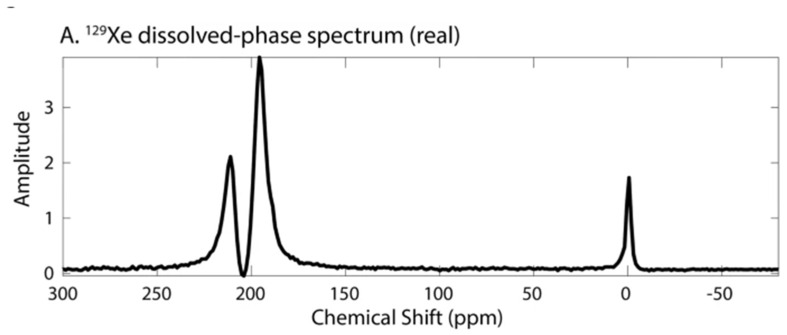
A HXe spectrum acquired in a healthy rat shows three distinct peaks, corresponding to gas in the alveoli/airways (set to 0 ppm), dissolved in tissue and blood plasma (197 ppm) and transiently bound to hemoglobin (~210.5 ppm). In vivo spectra in rats, humans, sheep and goats display distinct dissolved peaks that can be interrogated separately with an appropriate acquisition sequence, while those in pigs, rabbits and mice do not. Because the RF pulse was selective for dissolved components, the peak sizes greatly overrepresent the dissolved fraction.

**Figure 15 molecules-27-08338-f015:**
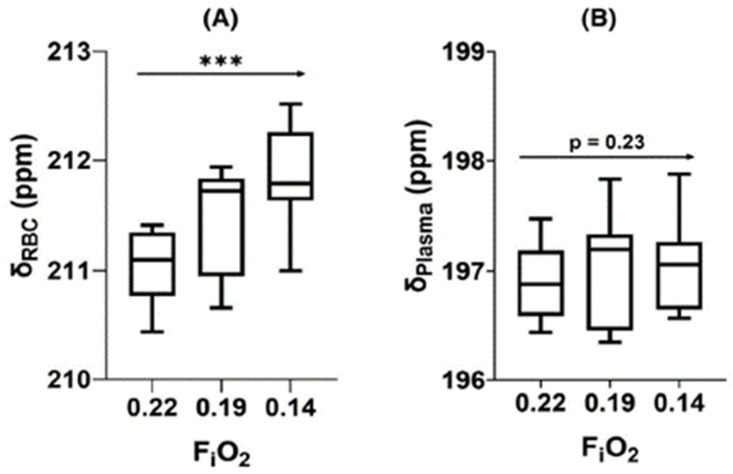
Measurements of ^129^Xe chemical shift in vascular blood in a healthy rat shows an increase in the chemical shift of the RBC peak as the FiO2 is lowered and hemoglobin oxygen saturation is reduced. This trend is opposite of that observed in human blood [57] but is consistent with the frequency shift observed between oxygenated and deoxygenated mouse blood in vivo, despite the unresolved RBC peak. *** = *p* < 0.001. Reprinted with permission from Friedlander, et al. (*Magn Reson Med* 2021, ref. [58]).

**Figure 16 molecules-27-08338-f016:**
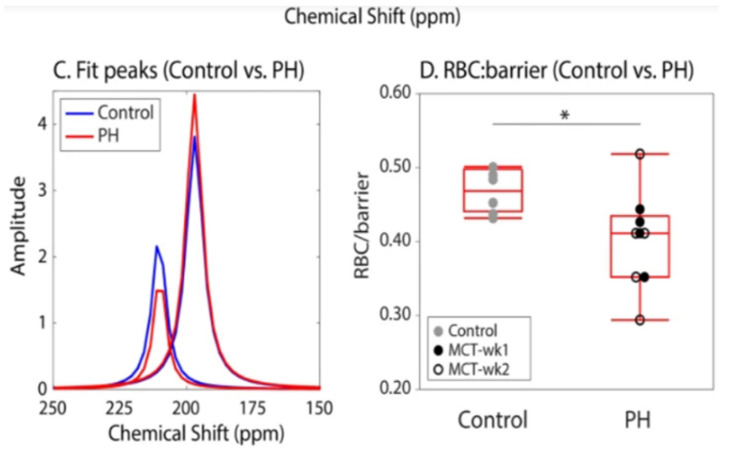
Instillation of a monocrotaline model in experimental rats reduces the observed ratio of RBC to barrier signals after one and two weeks. * = *p* < 0.05. Reprinted with permission from Virgincar, et al. (*Sci Rep* 2020, ref. [20]).

**Figure 17 molecules-27-08338-f017:**
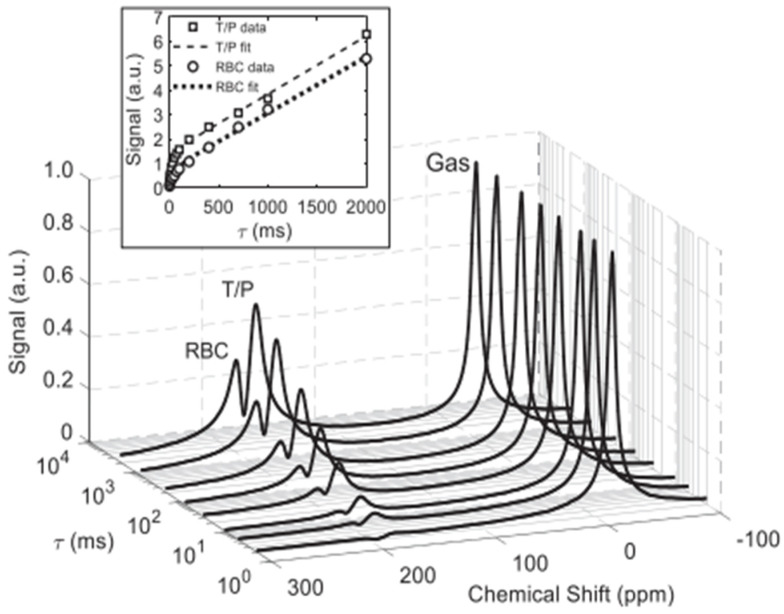
Spectra acquired after varying delay times show the gradual increase in amplitude of both membrane and RBC peaks as alveolar gas diffuses into those compartments and saturates in vascular blood. As seen in the inset, the two processes are well described as an approximately-exponential saturation of pulmonary blood and tissue, followed by an approximately-linear increase as the pulmonary blood is carried back to the heart and beyond.

**Figure 18 molecules-27-08338-f018:**
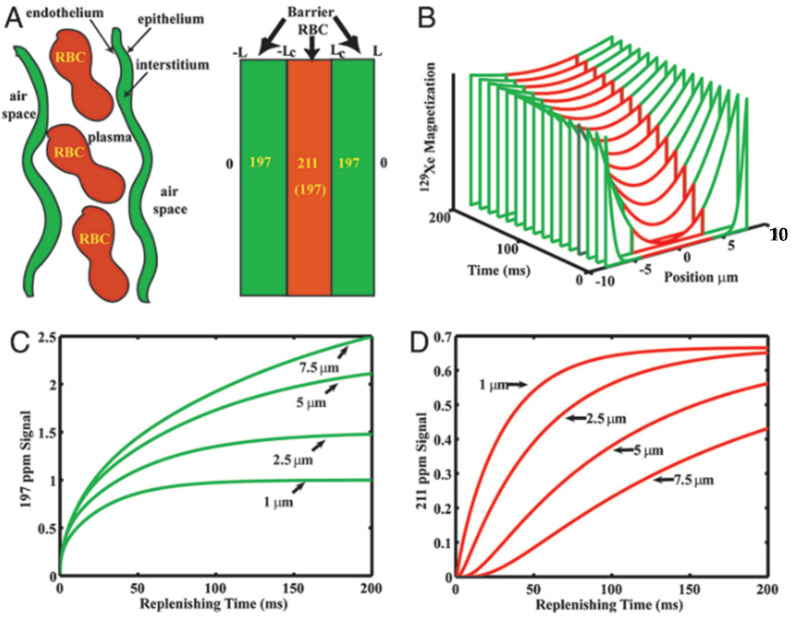
A simple model of alveolar wall structure (**A**) allows calculation of the expected diffusion dynamics (**B**) and the consequent temporal dynamics of the individual compartment peaks. Selection of the parameters that best describe observed behavior allows reframing in terms of microstructural dimensions of interest (**C**,**D**). Reprinted with permission from Driehuys, et al. (*Proc Natl Acad Sci USA* 2006, ref. [66]).

**Figure 19 molecules-27-08338-f019:**
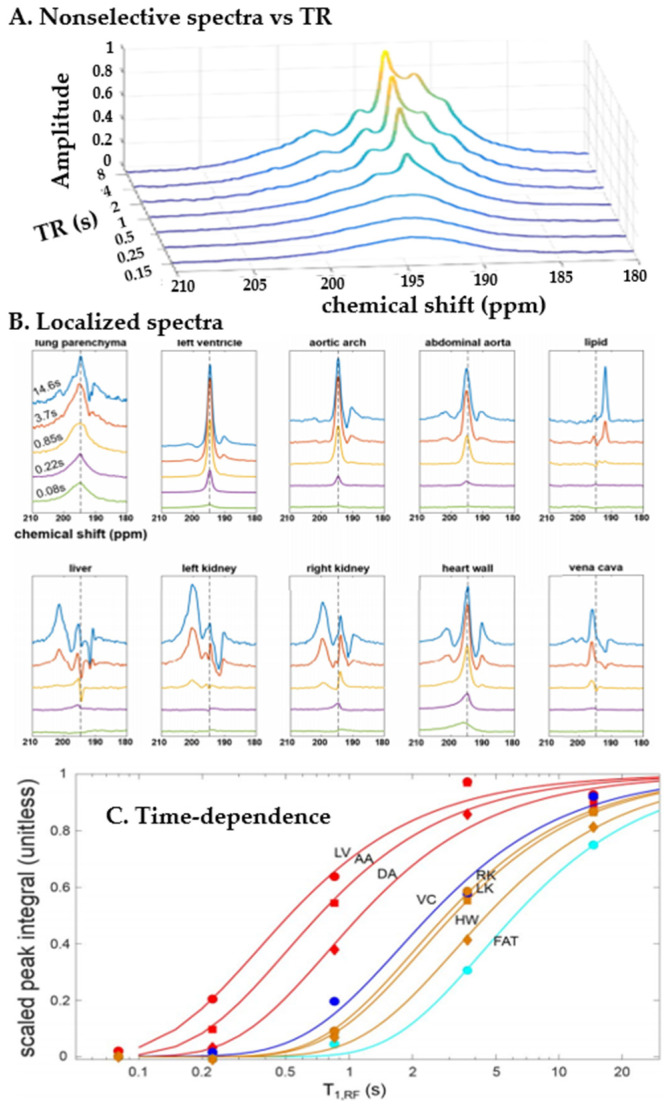
(**A**) Nonselective spectra acquired in a mouse as inhaled HXe is distributes throughout the body for progressively longer times. An initially broad, featureless peak is joined by five other distinct contributions over a period of ~8 s. (**B**) Localizing spectra to different anatomical regions shows that each has a distinct signature, largely due to locally different combinations of water, proteins and lipids, but perhaps also due to local magnetic susceptibility variations. (**C**) Localized dynamics of signal growth is also distinctive, and highlights how long it takes for pulmonary blood to transport dissolved gases to specific anatomical structures (LV: left ventricle, AA: aortic arch, DA: descending aorta, VC: vena cava, LK: left kidney, RK: right kidney, HW: heart wall, FAT: dorsal adipose deposits). This ranges from <<1 s for oxygenated blood to reach the heart, to >>5 s for gas-exchange to substantially affect dissolved concentrations in lipid-rich tissues on the animal’s back. Reprinted with permission from Loza, et al. (*IEEE Trans Med Imaging* 2019, ref. [59]).

**Figure 20 molecules-27-08338-f020:**
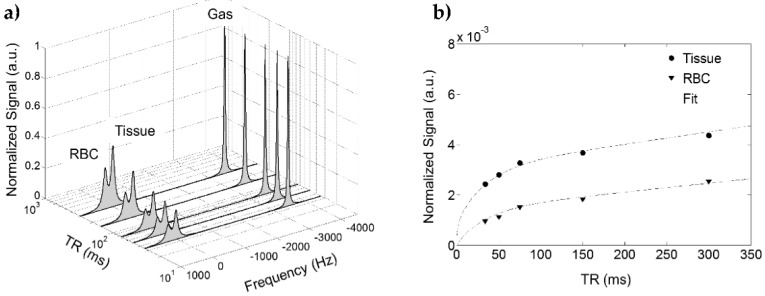
(**a**) Representative spectra acquired in a rat at various CSSR delay times and (**b**) the CSSR associated uptake curve. Reprinted with permission from Zanette, et al. (*Med Phys* 2018, ref. [75]).

**Figure 21 molecules-27-08338-f021:**
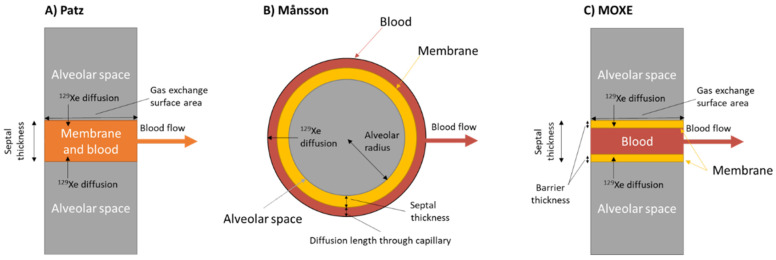
Different 1D analytical models of diffusion in the lung. (**A**) Shows the Patz model wherein xenon diffusion is modeled to diffuse from alveolar space into a combined membrane and blood compartment [76,77]. (**B**) Shows the Månsson that models the alveolus as a symmetrical sphere but considers the membrane and blood compartments separately [78]. (**C**) Shows the MOXE model that considers the same geometry as the Patz model but solves for both the membrane and blood signal [79].

**Figure 22 molecules-27-08338-f022:**
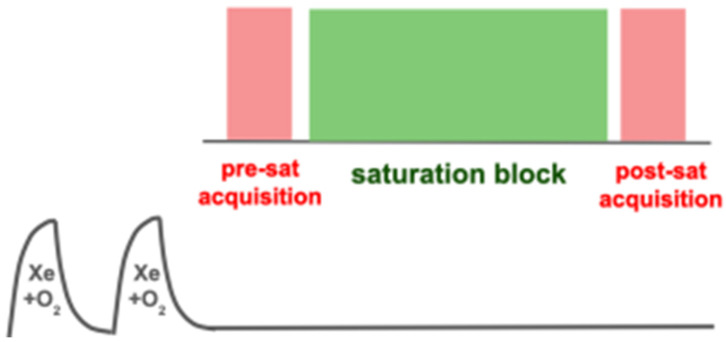
A general scheme for XTC measurements consists of one or more breaths during which HXe gas is introduced, and two MRI acquisitions separated by a period of ‘saturation’ (dissolved phase signal destruction). The extent of that destruction and its communication to the gas phase through diffusion is quantified using the difference between pre-sat and post-sat measurements.

**Figure 23 molecules-27-08338-f023:**
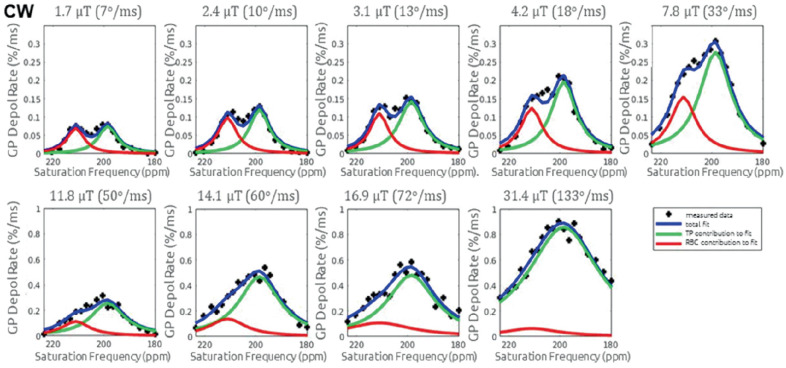
Z-spectra of the DP regime normalized by the RF pulse duration for saturation flip angle rates ranging from 7°/ms to 133°/ms. The spectra were fitted assuming two Lorentzian lines representing the depolarization rate from RBC and membrane compartments. At large applied power, it becomes impossible to separate out the effects of the saturation pulse on the two compartments. For example for RF intensities exceeding 60 deg/ms, depolarization in the tissue/plasma compartment (green curve) exceeds that of the RBC compartment (red curve) at all frequencies, rendering the RBC compartment effectively invisible. Note that, for the purpose of clarity, the top and bottom rows use different vertical scales. Reprinted with permission from Achekzai, et al. (*Magn Reson Med* 2022, ref. [103]).

**Figure 24 molecules-27-08338-f024:**
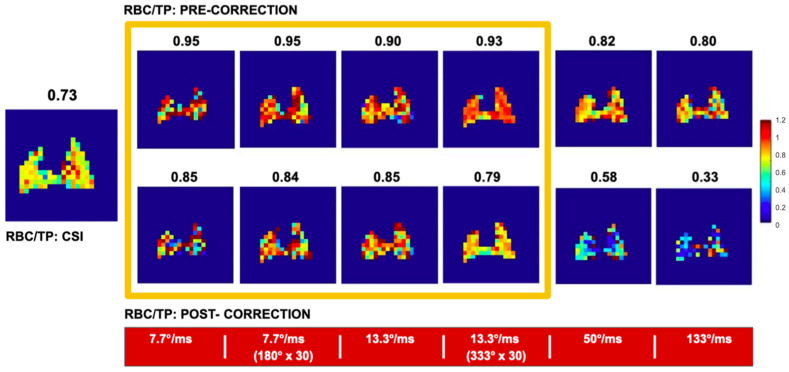
RBC/TP ratio maps compared to approximately equivalent ratio maps based on XTC MRI measurements using six different saturation schemes with average flip angle rates ranging from 7.7°/ms to 133°/ms, with and without correction for off-resonance excitation. Saturation schemes are either continuous, and specified by the flip-angle per ms, or discrete, specified by both average flip-angle per ms and the flip-angle x the number of saturation pulses applied (bottom red bar). Mean ratios consistently decrease to approach the ‘true’ value observed using direct Chemical Shift Imaging of the dissolved phase (left) and increase in fidelity as signal-to-noise is increased (left to right inside yellow box), but ultimately fail at the highest saturation power due to the off-resonance effects explained in Figure 23. Reprinted with permission from Achekzai, et al. (*Magn Reson Med* 2022, ref. [103]).

**Figure 25 molecules-27-08338-f025:**
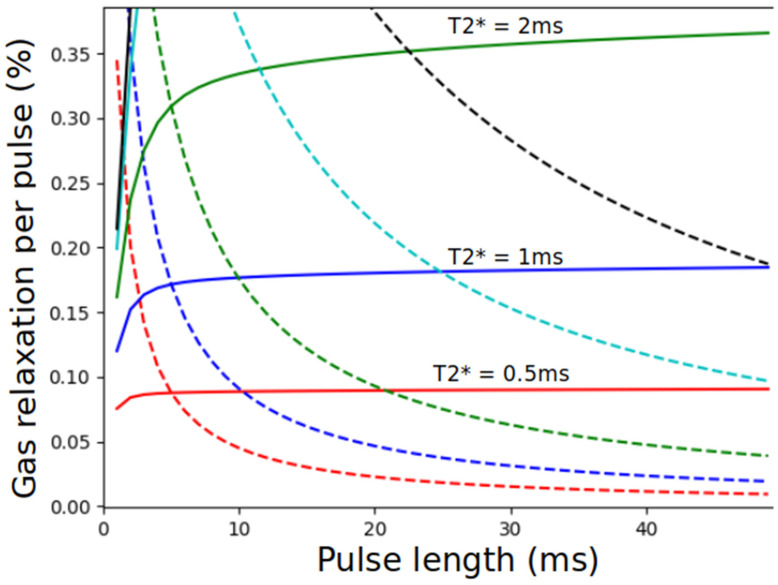
A Bloch simulation of the response to XTC (saturation) pulses shows that the extent of gas phase relaxation is largely independent of pulse length for pulses significantly exceeding the dissolved component T2* if the average pulse train power is held constant (solid traces) but not if the total flip angle is held constant (dashed traces). The figure is zoomed in to highlight behavior for the range of T2* (0.5–2 ms) likely to be encountered in clinical or preclinical imaging studies from 1.5 T to 9.4 T.

**Figure 26 molecules-27-08338-f026:**
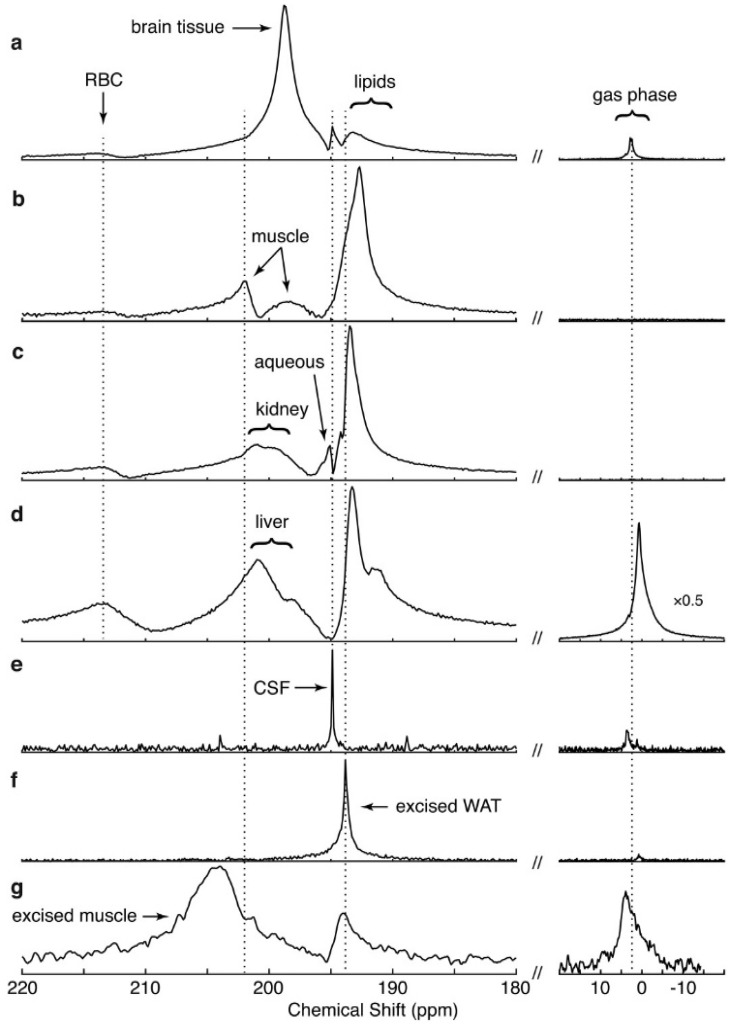
In vivo HXe spectra dissolved in the rat (**a**) head, (**b**) leg muscle, (**c**) kidney, and (**d**) liver. In vitro ^129^Xe spectra dissolved in rat (**e**) cerebral spinal fluid, (**f**) excised white adipose tissue, and (**g**) excised rat muscle. Reprinted with permission from Antonacci, et al. (*Magn Reson Med* 2018, ref. [105]).

**Figure 27 molecules-27-08338-f027:**
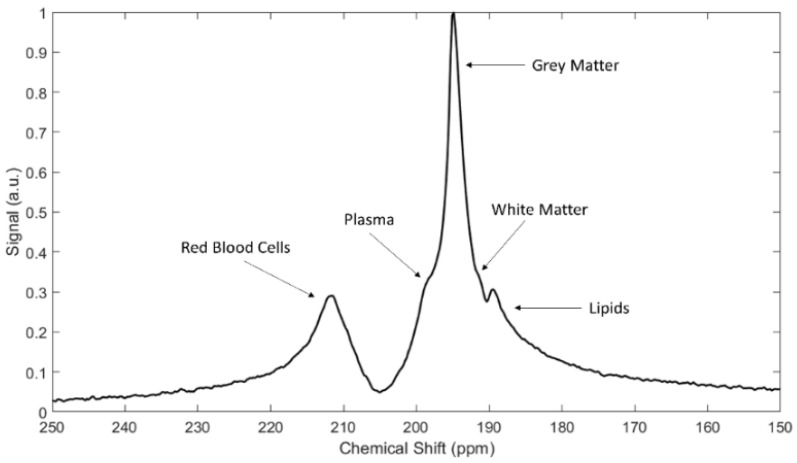
Representative HXe spectrum acquired in the rat brain. Reprinted with permission from Friedlander, et al. (*Magn Reson Med* 2021, ref. [106]).

**Figure 28 molecules-27-08338-f028:**
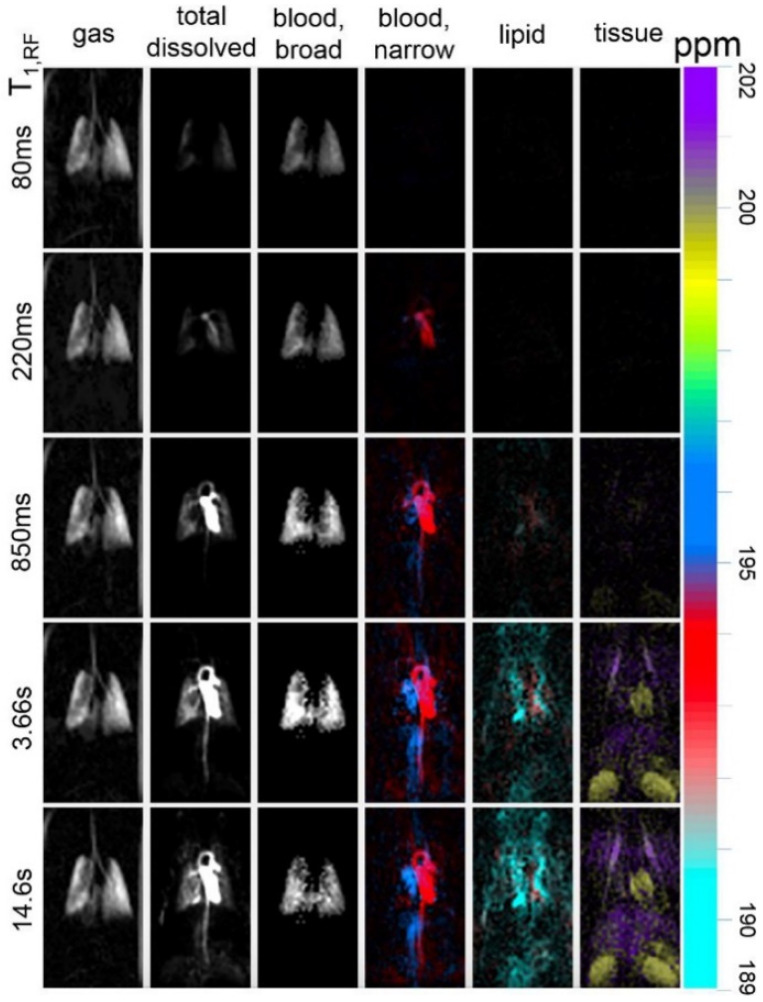
Distinct “compartmental” features along the circulatory system can be visualized by integrating under respective dissolved-phase linewidth peaks. *T*_1,_*_RF_* times (left) correspond to how long HXe magnetization persists in the body and, consequently, how far within the body it travels. The large range in chemical shifts associated with dissolved-phase HXe entering distinct anatomical compartments is visualized in individual, color-coded maps. Reprinted with permission from Loza, et al. (*IEEE Trans Med Imaging* 2019, ref. [59]).

**Figure 29 molecules-27-08338-f029:**
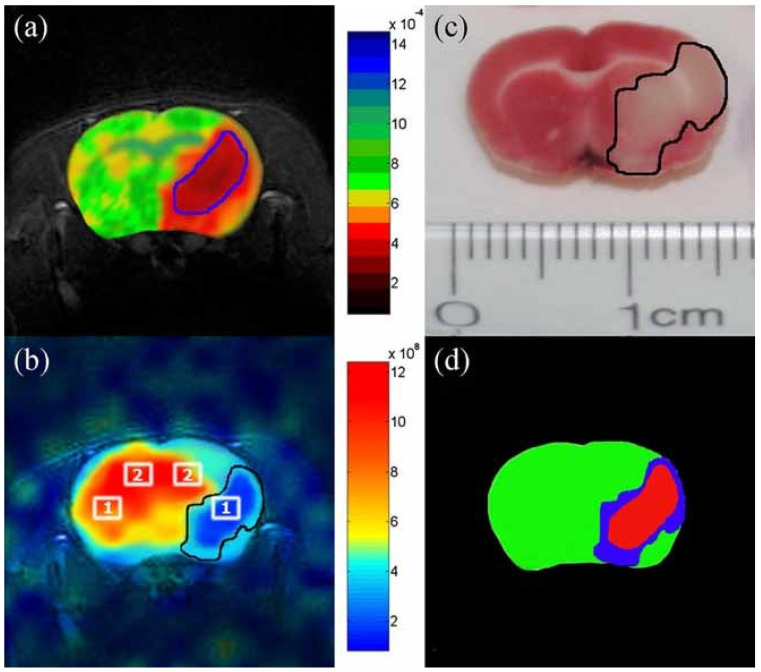
(**a**) Representative proton apparent diffusion coefficient (ADC) map image obtained 90 min after right middle cerebral artery occlusion (MCAO). The ischemic core is indicated by ADC values below 5.3 × 10^−4^ mm^2^/s (circled by a blue line). (**b**) Corresponding hyperpolarized ^129^Xe chemical shift image (CSI). There is a large signal void in the ipsilesional (right) hemisphere. The defined regions of interest (ROIs) are labeled as follows: ROI1, core; ROI2, normal tissue. The xenon signal intensity is given in arbitrary units. (**c**) Corresponding 2,3,5-triphenyltetrazolium chloride (TTC)-stained brain section of the same animal as in (**a**) and (**b**). (**d**) Tricolor map based on the ADC and TTC images shown in (**a**) and (**c**). Green, red and blue represent nonischemic tissue, core and penumbra, respectively. Reprinted with permission from Zhou, et al. (*NMR Biomed* 2011, ref. [109]).

**Figure 30 molecules-27-08338-f030:**
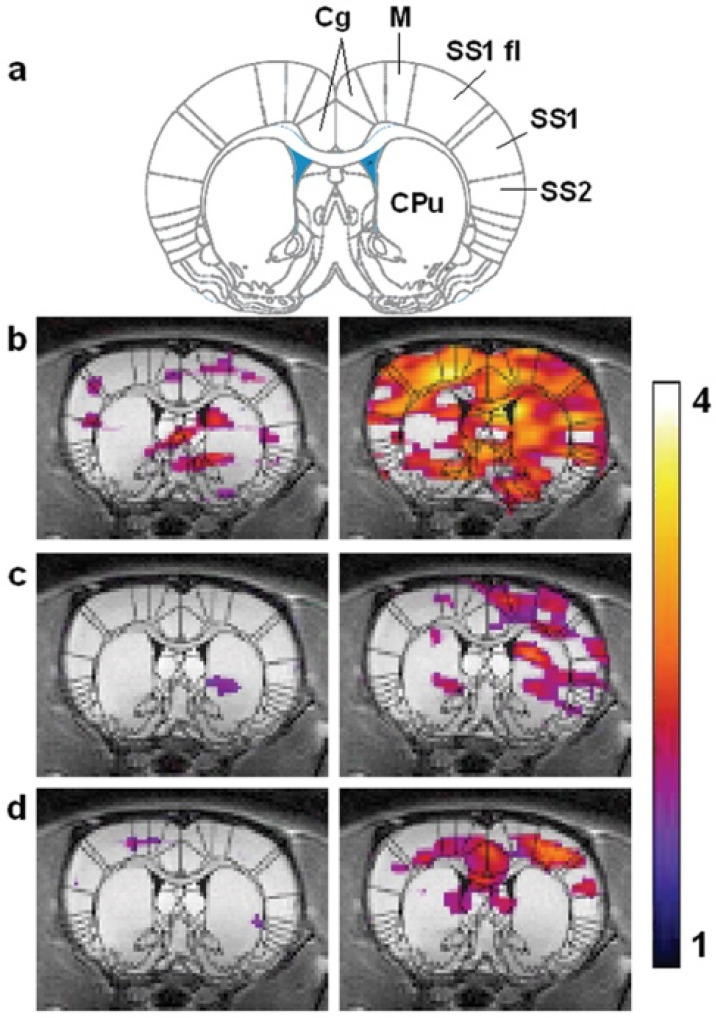
The HXe signal is shown as a false colour overlay on the corresponding 1 mm thick coronal proton reference image taken from the same animal. The left panel shows HXe signal intensity during baseline and the right panel shows HXe signal intensity after injection of capsaicin 20 uL (3 mg/mL) into the right forepaw. Colour scale represents SNR and only signal with SNR above 2 are shown. Superimposition of a rat brain atlas demarcates specific areas of the brain: cingulate cortex (Cg), motor cortex (M), primary somatosensory cortex and SS1 forelimb region (SS1 and SS1 fl), secondary samatosensory cortex (SS2), and striatum (CPu). Reprinted with permission from Mazzanti, et al. (*PLoS ONE* 2011, ref. [121]).

**Figure 31 molecules-27-08338-f031:**
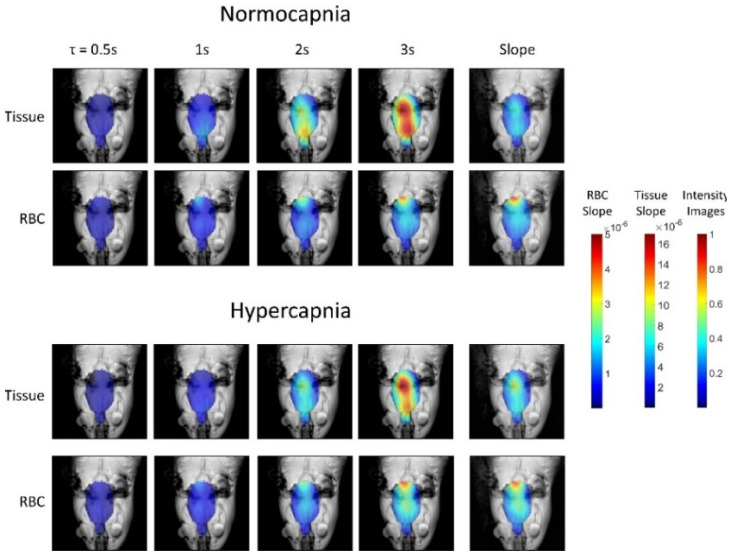
Spectrally resolved of HXe dissolved in the rat brain at various washout times during normocapnic (top) and hypercapnic (bottom) ventilation. Reprinted with permission from Friedlander, et al. (*Magn Reson Med* 2021, ref. [106]).

**Figure 32 molecules-27-08338-f032:**
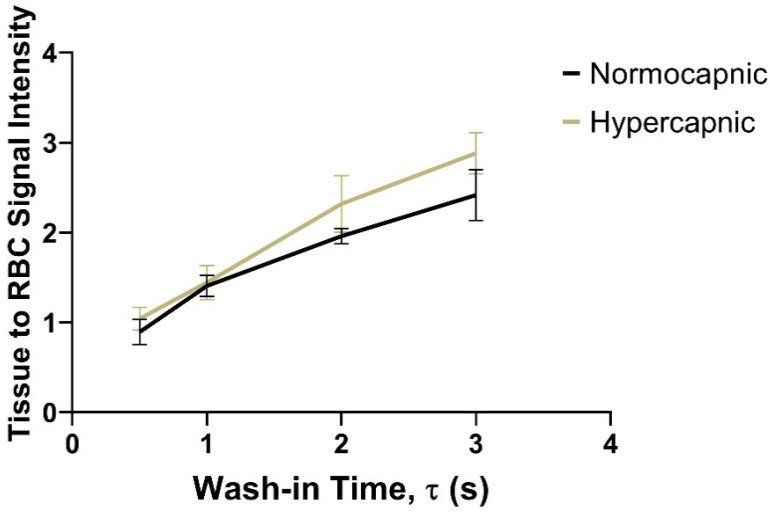
The ratio of tissue to RBC signal intensity in the rat brain at various wash-in times. The experiments demonstrate increased gas-transfer during hypercapnia. Reprinted with permission from Friedlander, et al. (*Magn Reson Med* 2021, ref. [106]).

**Figure 33 molecules-27-08338-f033:**
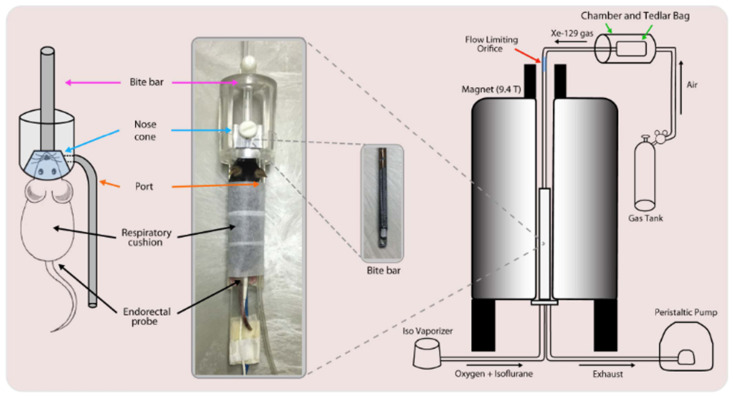
(**Left**) Mice were held in an animal cradle consisting of a bite bar to adjust the animal’s position, and a nose cose through which gas is supplied. The port was used as an exhaust for exhaled gas. (**Right**) O_2_, air, and isoflurane were supplied from beneath the bore, while HXe was supplied from above through the bite bar: the gases mix within the nose cone, supplying the animal with a normoxic gas micture. Exhaled gas and isoflurane were passed through a peristaltic pump and isoflurane scrubber. Reprinted with permission from Loza, et al. (*IEEE Trans Med Imaging* 2019, ref. [57]).

**Figure 34 molecules-27-08338-f034:**
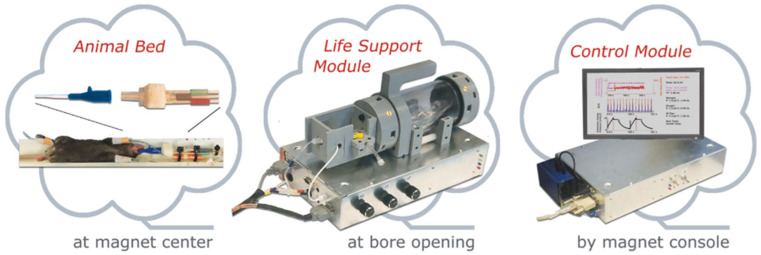
Primary subsystems of an HP gas rodent ventilator. At the magnet center, the rodent lies on the animal bed, connected to physiologic monitoring sensors. The intubated animal (the intubation catheter is magnified) is linked to the life support module, which resides at the entry of the bore and contains the gas-delivery valves, physiologic monitoring circuitry, and HP gas supply. The life support module is connected via a single cable to the control module, which is located next to the MRI user console. Reprinted with permission from Virgincar, et al. (*J Magn Reson* 2018, ref. [127]).

**Figure 35 molecules-27-08338-f035:**
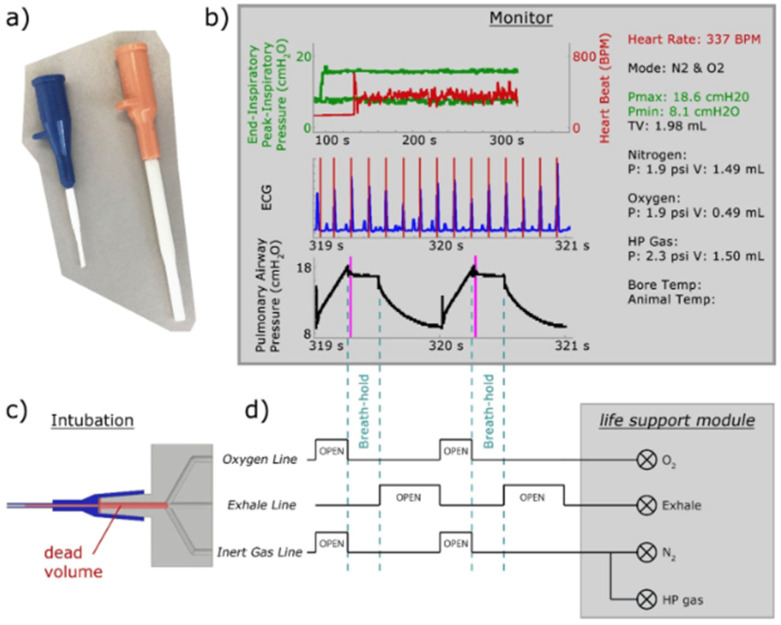
(**a**) Tapered intubation catheters for mice (22 G, blue) and rats (14 G, orange). (**b**) Display of the ventilator while ventilating a rat, showing real-time airway pressure (black), the MR trigger (pink), ECG (blue) with automatic r-wave detection. The evolution of the peak inspiratory and expiratory pressures (green), and heart rate (red) over the duration of the study are also shown. The top right corner of the display shows the current heart rate, ventilation mode, airway pressures, gas-delivery pressures and associated tidal volumes. (**c**) Cross-sectional view of the intubation catheter connected to the mouthpiece, showing the merging gas lines, and the dead volume. (**d**) Actuation pattern of the gas delivery valves (drawn in synchrony with the airway pressure trace on the monitor). Reprinted with permission from Virgincar, et al. (*J Magn Reson* 2018, ref. [127]).

**Figure 36 molecules-27-08338-f036:**
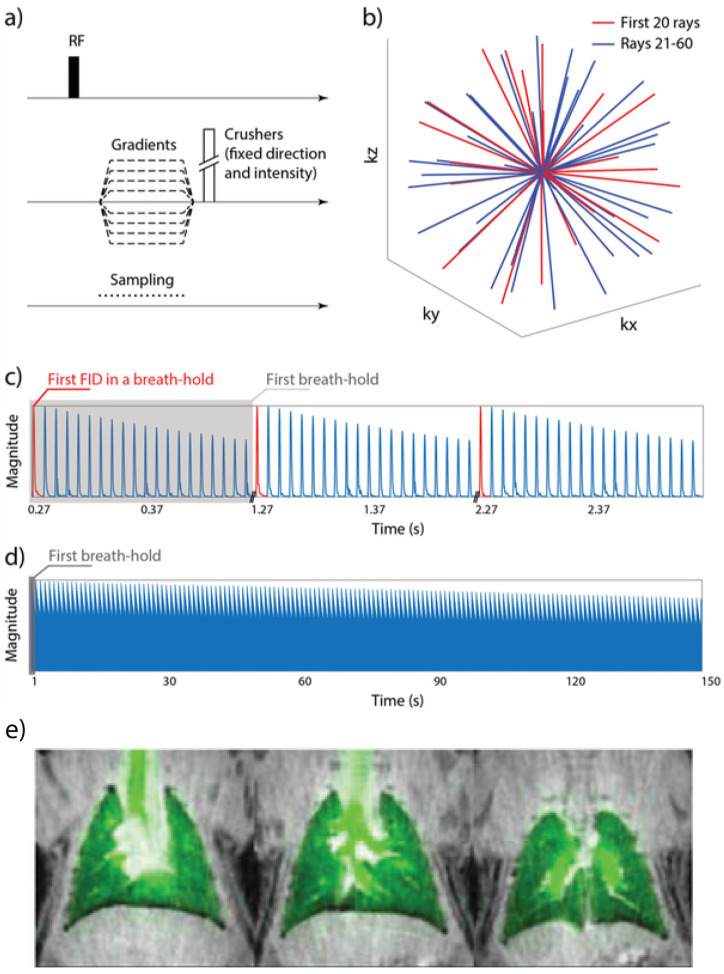
(**a**) Schematic of the UTE3D imaging sequence. (**b**) 60 representative radial rays, acquired in randomized order, to sample k-space uniformly over time (illustrated by the first 20 rays being shown in red). (**c**) Raw HXe gas-phase FIDs from the first three breath-holds of a rat imaging study. Within each breath-hold, 20 FIDS are acquired, with the amplitude of each subsequent FID decreasing due to RF-induced loss of magnetization. However, this is replenished by continued ventilation, and the resulting HXe signal intensity remains stable over the course of the acquisition, with only a slow decrease due to T1 relaxation in the dose bag (**d**). Note that the time axis in (**c**) is discontinuous because no data was recorded between breath-holds. (**e**) Resulting HXe gas-phase image. Reprinted with permission from Virgincar, et al. (*J Magn Reson* 2018, ref. [127]).

**Figure 37 molecules-27-08338-f037:**
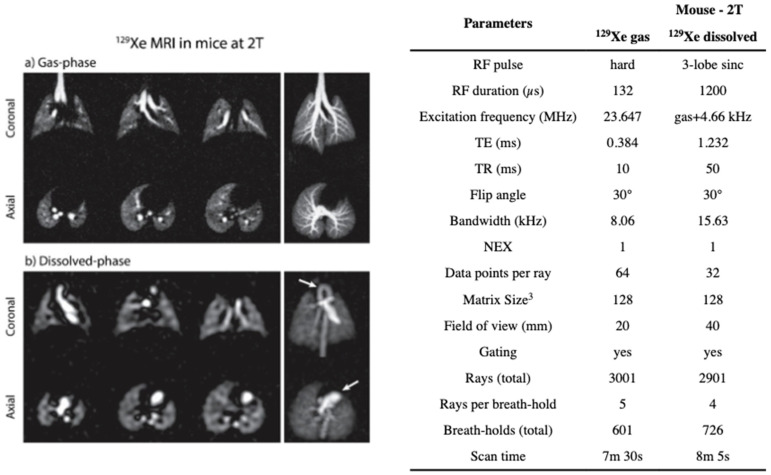
Gas-phase and dissolved-phase images in a mouse in vivo at 2 T. The white arrows denote visualized anatomical structures using dissolved HXe (aortic arch, top, and left ventricle, bottom). Reprinted with permission from Virgincar, et al. (*J. Magn. Reson.* 2018, ref. [127]).

**Figure 38 molecules-27-08338-f038:**
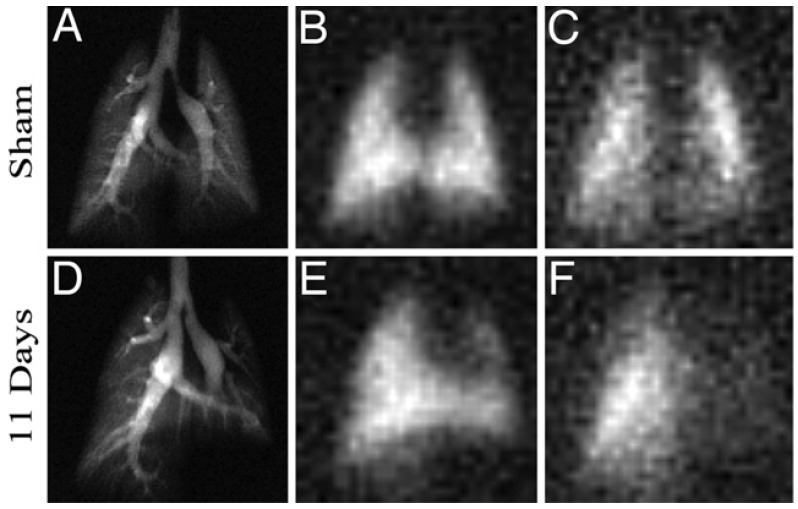
Comparison of HXe images in sham left-lung instilled animal (**A**–**C**) vs. injured animal with left-lung fibrosis 11 days postinstillation (**D**–**F**). (**A**) Airspace image in sham. (**B**) Barrier image in sham. (**C**) RBC image in sham. (**D**) Airspace image in injured animal. (**E**) Barrier image in injured animal. (**F**) RBC image in injured animal. Most notable is the nearly complete absence of HXe-RBC intensity in the injured left lung of the diseased animal (**F**), indicating that HXe does not reach the RBCs on the 50-ms image acquisition time scale, likely resulting from increased diffusion barrier thickness. However, note that the barrier images (**B**) and (**E**) closely match the corresponding airspace images (**A**) and (**D**). Reprinted with permission from Driehuys, et al. (*Proc Natl Acad Sci USA* 2006, ref. [66]).

**Figure 39 molecules-27-08338-f039:**
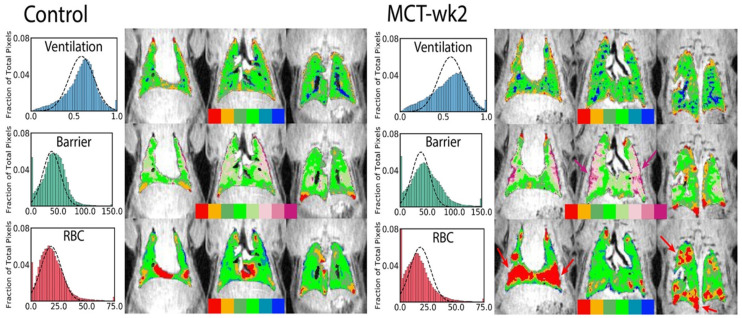
Representative ^129^Xe ventilation, barrier:gas, and RBC:gas maps with histograms in healthy and pulmonary hypertension (PH) rats. The dotted line in the histogram represents the reference distribution derived from healthy animals. The maps from the control group show homogeneous ventilation, barrier-uptake, and RBC-transfer signal with a few baseline defects in barrier and RBC signal, mostly confined to the base of the lung. Gas-exchange maps in PH models show reduced HXe uptake in RBCs. Additionally, enhanced barrier signal and a modest increase in ventilation defects were also observed. Adapted with permission from Virgincar, et al. (*Sci Rep* 2020, ref. [20]).

**Table 1 molecules-27-08338-t001:** A summary of gas exchange experiments performed in animals.

Paper	Model	Species	Disease Model	Statistically Significant Outcome Measurements
Imai et al. (2010) [80]	Patz	Mice	Emphysema	SVR
Freeman et al. (2013) [81]	MOXE	Mice	--	--
Fox et al. (2014) [82]	Månsson	Rats	Radiation induced lung injury	T_TR_Tissue_ ^†^
Li et al. (2016) [83]	MOXE	Rats	Radiation induced lung injury	Capillary transit time, septal wall thickness, haematocrit
Zhang et al. (2016) [83]	MOXE	Rats	Cigarette smoke injury	Septal wall thickness, air-capillary barrier thickness
Doganay et al. (2016) [84]	Månsson	Rats	Radiation induced lung injury	Pulmonary tissue thickness, relative blood volume
Zhong et al. (2017) [74]	MOXE	Rats	--	--
Zanette et al. (2018) [75]	MOXE	Rats	Radiation induced lung injury	Air-capillary barrier thickness, haematocrit
Zhang et al. (2020) [85]	MOXE	Rats	Fine particulate matter pollution	Septal wall thickness
Ruppert et al. (2020) [86]	Patz	Rabbits	--	--
Fliss et al. (2021) [87]	MOXE	Rats	Bronchopulmonary dysplasia	Mean chord length, SVR, haematocrit

† The authors did not report the calculated morphometric measurements from the model but rather reported only the time constants.

**Table 2 molecules-27-08338-t002:** Relative ^1^H and HXe magnetization available for MRI. γ_P_ refers to the gyromagnetic ratio of the proton. These approximations assume that 80% of alveolar gas is replaced with 85% ^129^Xe-enriched HXe.

	Nuclear Density	Polarization (%)	Gyromagnetic Ratio	Magnetization @ 3 T
**^1^H (conventional MRI)**	30 M	3.3 × 10^−4^ × (B/T)	γ_P_	1
**HXe Gas phase**	0.03 M	~20	0.28 γ_P_	9
**HXe Dissolved phase**	0.0006 M	~20	0.28 γ_P_	0.18

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
