# Peer review of "Preclinical MRI Using Hyperpolarized ^129^Xe"

_molecules, 2022, doi:10.3390/molecules27238338_

Round 1
Author Response
Reviewer 1:
Kadlecek and co-authors present a comprehensive review of the motivation, challenges, and desired outcomes of hyperpolarized 129Xe translation from preclinical animal imaging to clinical human imaging. The literature reviewed spans a range of foci including imaging study design and analytical modelling, as well as spectral and image analysis in and ex vivo, and is presented in a concise and logical format. Recent reviews in the field of preclinical and clinical 129Xe hyperpolarization have focused on human brain imaging (Shepelytskyi et al. 2022) and underlying physics of contrast agent production (Khan et al. 2021), so this review fulfils an unmet need in complement to existing work. The review of recent underlying literature is thorough with no notable omissions or excessive self-citation. The conclusions clearly summarise the current “state-of-play” and provide sensible hypotheses for future work. I have a small number of technical questions and would like to suggest some minor changes that I believe would improve the readability of the manuscript, but otherwise I otherwise recommend the manuscript to be accepted.
- Gas dynamics during the breathing cycle P3 L5 – State the acronym (IPA) here for easier reference to the figure and caption, which is currently displayed on the previous page. P3 Fig. 1.2 – Is it to be assumed that the difference between dark and light green voxels is: dark = “μ - σ < I < mean” and light = “mean < I < μ + σ” on L38? Similarly, the light and dark blue overventilated ranges are not clearly defined in the same way that ventilation defect regions are – should dark blue represent I > μ + 2σ, for example?
The acronym IPA stands for Invasive Pulmonary Aspergillosis. This is now noted in the manuscript.
Regarding the threshold for the different color bins, the reviewer’s assumption is correct. We have updated the text to make this more clear:
“Voxels that are determined to be within the lung boundaries and outside of major airways (based on comparison to 1H images) are classified as defect (red, intensity I < μ - 2σ of healthy distribution), partial defect (orange, μ - 2σ < I < μ - σ), normal (dark green, μ - σ < I < μ and light green, μ < I < μ + σ), hyperventilated (light blue, μ + σ < I < μ + 2σ and dark blue, I > μ + 2σ).”
- P5 Fig. 1.6 – What is the significance of leading “H” markers in “HXe” and “HHe” here?
The leading H means ‘Hyperpolarized’. We have modified the text to explain it when first introduced and use it consistently throughout.
- Diffusive dynamics in the lung parenchyma P8 Fig. 2.4, 2.5 – Might be an issue with formatting or conversion during the article submission process, but recommend checking the quality of these images as they came across as quite grainy in the review copy and important information is hard to discern.
We have replaced figures with those of high quality (>= 320 DPI) throughout. We were (and are) uncertain about how figures may be changed when imported into Microsoft Word, or if they should be inserted by the publisher separately but all figures will be easily readable in the final publication.
- Spectra and Compartments P10 Fig 4.1 – You mention that dissolved Xe spectral peaks in pigs, rabbits and mice are not resolvable – does this refer to a body of work in the literature? This is somewhat alluded to around L31 of this page, but no detailed context is present in the figure caption, and neither of the references 39-40 included in the main text describe animal studies.
The reviewer brings up an important point that we were unsure how to handle. We, and likely others, have done in vivo and in vitro experiments in a variety of species. The statement in the manuscript is generally agreed to be true, but there are some persistent inconsistencies that have prevented publication. In particular, the spectral separation in blood samples does not always reflect that observed in vivo, and to the best of our knowledge there is no explanation for this yet.
In an initial draft, we had included our in vitro spectra from blood samples of rat, pig (two strains), goat and sheep, all of which show two distinct peaks. However, we find that in vivo pigs show only minimal separation, and only at long echo times, and the separation between the peaks is approximately halved (10ppm vs 20) in rats. Since this may arise from some unknown technical problem we didn’t want to mislead the reader by including it, but at the same time we thought it important to note that as a practical matter separate interrogation of the RBC compartment in mice, pigs and rabbits appears to be impossible. We have modified the text to briefly discuss this ongoing mystery and have decided not to include the spectral measurements here or anywhere else until we understand them better.
- P12 L8 – Text refers to Figures 3.4 and 3.1, but all figures in this section are 4.X, so please check that the main body text and figure labels are consistent (Figures 3.X seem to have been skipped).
The reviewer is correct, we reorganized the sections at the last minute for what we thought was better flow and missed these figure references in the text. As per reviewer 2’s suggestion, we have modified section order again and have modified figure and subsection numbers accordingly.
- P12 Fig. “4.5” – The text describes six distinct peaks / features in the mouse model, and whilst the reader can find out more information from this in the associated reference 44, it would be nice to state what these different features are here, since this information is not present in the reprinted figure. Additionally, I believe this figure should be 4.6, as the alveolar wall model on the previous page is already assigned Fig. 4.5.
Same figure numbering error, now corrected. As for the six peaks, we had seen six distinct bumps in the spectra that when localized corresponded primarily to pulmonary blood/tissue (spectrally broad), vascular oxygenated blood, vascular deoxygenated blood (both spectrally narrow), lipid, heart wall + kidneys, liver and muscle. There is almost certainly some overlap between these compartments, and other organs or locations may display additional peaks, particularly the multiple peaks seen in brain, which was not in the sensitive area of our coil. This is all a little complicated, which is why we didn’t get into it, but we agree that it was not specific enough in the text and have included three new sentences to describe as above.
- Practical Imaging Considerations 2 P14 L37 – Is it not redundant to state that ventilator designs need to be both non-ferrous and MR compatible? (unless localised heating becomes a factor I suppose…) Additionally, I believe most safety literature now uses the term “MR conditional” instead of “compatible”?
We have changed to conditional rather than compatible as suggested. The term non-ferrous was vague. We meant to point out that some materials that display no bulk magnetism, like aluminum or brass, still have sufficient ferromagnetic surface impurities to quickly relax HP gas, so direct contact with the gas should be avoided. The language has been clarified.
- P16 Table 5.4 – Please include units for the Polarization (%) in the column header. Could also specify that you’re using relative magnetization (so units are not required) in the last column header, but it’s also specified in the figure caption, so this isn’t essential.
Done as suggested.
- P16 L20 – Explain that depolarization of Xe following contact with oxygen is due to paramagnetic effects if it’s not otherwise explained elsewhere.
We thank the reviewer for this suggestion. The text has been updated to clarify the paramagnetic effect:
“inhaled xenon gas comes in contact with paramagnetic oxygen, which accelerates its T1 relaxation time from several minutes outside the body to ~19 s in the lungs”.
- “Xenon Polarization Transfer Contrast” (XTC)-based techniques P26 Fig 7.4 – Is it intended that some of the long T2* curves are not shown due to the chosen scale of the y-axis? P27 L4 states that short pulses are undesirable, so it seems odd that the data for short pulses is in focus whilst the data for long pulses is not entirely visible. Also, please include the pulse length as an x-axis label.
Because the T2* of the dissolved components is short at preclinical imaging fields (~2ms at 1.5T, reduced to ~0.67ms at 9.4T in a mouse model), it is unlikely that any experimenter will be interested in behavior outside that range. We have explained that in the figure caption. We failed to note that the x label was obscured by the figure caption and have fixed that error.
Reviewer 2 Report
The authors provided a comprehensive review of preclinical hyperpolarized (HP) xenon-129 (129Xe) MRI studies. This manuscript covers all major spectroscopic and imaging techniques for pulmonary imaging as well as whole-body and brain imaging. The manuscript is important for the field of HP 129Xe MRI, however, requires some major changes and restructuration prior to publication. Please find my general and specific comments below:
General Comments
1. First of all, the structure of the manuscript has to be changed. Currently, the manuscript jumps back and forth between spectroscopy and imaging making the reading somewhat uneven and inconsistent. I suggest moving the description of spectral components (Section 4) of HP 129Xe right after the ventilation image discussion (Section 2). Once spectroscopy is discussed, I suggest including gas exchange models since they explain the origin of spectral peaks. Dynamic spectroscopy might be moved as well as a natural continuation of the gas exchange models. After that, I suggest putting Section 5 which includes gas-phase (Section 5.2.2) and dissolved-phase imaging (Section 5.2.3). Following dissolved-phase imaging, I would include XTC as another technique that allows obtaining similar information (Section 7). Next, diffusion imaging (Section 3) can be discussed. The resulting arrangement of the section in the manuscript (1-2-4-5-6-5-7-3-8) seems more logical to me and will have a better story flow.
2. What surprised me, is the complete absence of references in the introduction section. There are many statements in the introduction that should be referenced properly. For example, statements such as “Among imaging modalities, MRI is particularly well suited to this task because it derives from and can make use of NMR-based techniques that encode rapid dynamical behavior as changes in the frequency, linewidth or intensity of spectral features.”, “Lung disease and its diagnosis is therefore extremely challenging, requiring an understanding of the muscular and airflow aspects of breathing on the scale of seconds and centimeters, to the diffusive dynamics of gases in the sub-millimeter regime, down to the millisecond- and micron-scale transport of gases across capillary walls and in red blood cells.”, and “This includes 1H metabolite or fat/water imaging, or hyperpolarized 13C metabolic imaging.” should be referenced. The authors should go over the introduction and include proper citations throughout.
3. Some images seem to be low resolution and blurry. The authors should improve the visibility of the figures. Specifically, figures 2.4, 2.5, and 5.3.
4. In the Whole Body and Brain Imaging section, the authors should include recent preclinical achievements in HP 129Xe molecular imaging (results by McHugh CT., et. al. Magn Reson Imag. 2021; 87(3); 1480-1489 as well as Hane FT., et.al. Sci Rep. 2017; 7:41027). This field is unique for HP 129Xe MRI and recent preclinical success in hyperpolarized chemical exchange saturation transfer is of high interest for HP 129Xe MRI and molecular imaging.
Specific comments
1. Page 2, Introduction. “…breathing, heart rate and circulatory rates in rodent models are far higher, as are the available imaging magnetic fields and gradients.”
It should be mentioned as well that preclinical HP 129Xe imaging is also associated with technical challenges such as a requirement for the small dedicated coils, additional equipment such as an animal breathing ventilator, etc.
2. Page4. Section 2. Eqns. 1-2. Some dots appear in the equations at random places. For example, in Eqn. 2 a dot appeared on top of both minus signs. Eqn.1. has a dot on top of V in nominator in the first term. Is that a different volume than in the denominator? Please correct or explain the meaning of those dots.
3. Page 6. Section 3. While talking about the implementation of ADC gradients into spectroscopy, the authors may also reference work by Boudreau M., et. al. (Boudreau M., et. al. Magn Reson Med. 2013;69(1):211-220).
4. Page 11. Section 4. “Within approximately 100ms after inhalation, dissolved xenon accounts for ~0.2-4% of the total 129Xe signal, depending on species and disease state.”
The author should provide a reference for this statement.
5. Page 11. Section 4. “The limited amount of dissolved xenon, along with the short T2* characteristic of lung tissue …”.
The authors should provide a T2* value for HP 129Xe dissolved in the lung parenchyma.
6. Page 12. Section 4. Figure 4.5 looks like three figures of different sizes rather than a single figure. The authors should make sure, that plots A, B, and C are actually united into a single graphical object.
7. Page 12, Section 4. “…these peaks appear over a much longer timescale, as shown in Figure 3.5A.”.
I guess the authors meant Figure 4.5 here. Please double-check.
8. Page 13. Section 4. “…spectroscopy, as shown in Figure 3.5B”.
Again, I assume it should be Figure 4.5B
9. It seems that there is no reference to Figure 4.5C in the text. If there is no reference to it, the authors should remove Figure 4.5C from the manuscript. Alternatively, please reference Figure 4.5C where it is appropriate.
10. Page 15. Section 5.2.1. Was natural abundance (~26%) used for the calculation of HP 129Xe magnetization or isotopic enrichment of 129Xe (~85-90%) was assumed? Please specify.
11. Page 16. Section 5.2.1. Table 5.4. The authors should change the table to reflect polarization values used from different polarizers in different groups. The polarization column should reflect a practically achievable polarization range. For example, 15% polarization was used for preclinical imaging by McHugh CT., et.al. (McHugh CT., et. al. Magn Reson Imag. 2021; 87(3); 1480-1489), whereas the highest polarization reported for animal scans was ~30% (Chahal S., et. al. Brain imaging using hyperpolarized 129 Xe magnetic resonance imaging. In: Eckenhoff RG, Dmochowski IJ, eds. Methods in Enzymology. Vol 603. Academic Press Inc. 2018:305-320.). The obtained magnetization range for the 0.15-0.3 polarization range should be provided in the table (with appropriate references).
12. Page 16. Section 5.2.2. Figure 5.5. I suggest including the image acquired using the shown UTE3D pulse sequence in the Figure. This will allow the reader to understand the outcome image right away.
13. Page 16. Section 5.2.2. “… the transverse magnetization decays with a short T2* of ~5ms at 7 Tesla49”.
I suggest writing 7T to maintain consistency with the other manuscript sections.
13. Page 16. Section 5.2.2. “…as GRE, 3D radial (UTE3D), spiral,…”
The authors should include references for each technique mentioned.
14. Page 17. Section 5.2.2. “Typically, the scan time is limited to ~10 minutes…”
Reference is required here.
15. Page 17. Section 5.2.2. “Most commonly, a fixed small flip angle is used, which induces a modest amount of depolarization with each excitation.”
References are required for this statement.
16. Page 18. Section 5.2.4. “…technique, where a single echo time is employed, which corresponds to the two resonances being 90° out of phase…”.
The authors should include the value for TE.
17. Page 22. Section 6.3. Table 6.1. The authors gathered eleven studies in the table and mentioned them in the text. However, the main findings of those works were not highlighted. Based on Table 6.1 it seems that septal wall thickness and hematocrit are two of the most commonly used biological parameters for gas exchange studies. The author should at least describe disease-caused changes in these parameters. Providing the explanation along with equations for septal wall thickness and hematocrit calculation would be also helpful (and useful) for the reader.
18. Page 30. Section 8.3. The author mentioned Peled’s model only, which was historically the first model of the HP 129Xe signal in the brain. The authors should also talk about and discuss the other HP 129Xe uptake models such as Martin’s model (Martin CC., et. al. J Magn Reson Imag. 1997; 7:848-854), Kimura’s model (Kimura A., et. al. Magn Reson Med Sci. 2008; 7:179-185), and Imai’s model (Imai H., et. al. NMR Biomed. 2012; 25:210-217). I would encourage the authors to compare these models similarly to how they compared the models of HP 129Xe gas exchange in the lungs.
Author Response
Reviewer 2
- First of all, the structure of the manuscript has to be changed. Currently, the manuscript jumps back and forth between spectroscopy and imaging making the reading somewhat uneven and inconsistent. I suggest moving the description of spectral components (Section 4) of HP 129Xe right after the ventilation image discussion (Section 2). Once spectroscopy is discussed, I suggest including gas exchange models since they explain the origin of spectral peaks. Dynamic spectroscopy might be moved as well as a natural continuation of the gas exchange models. After that, I suggest putting Section 5 which includes gas-phase (Section 5.2.2) and dissolved-phase imaging (Section 5.2.3). Following dissolved-phase imaging, I would include XTC as another technique that allows obtaining similar information (Section 7). Next, diffusion imaging (Section 3) can be discussed. The resulting arrangement of the section in the manuscript (1-2-4-5-6-5-7-3-8) seems more logical to me and will have a better story flow.
We experimented with the suggested ordering, but in the end settled on the reordering (1-2-3-4-6-7-8-5). We appreciate the suggestion, as it made us think through and explain much more clearly why things are in the order that they are. Our thought was essentially to follow the path of the inhaled xenon through the airways, into the airspaces, dissolving into first tissue then binding to RBC, and finally being returned to the airspace or delivered to the rest of the body. We have completely redone the introduction to explain this organizational scheme.
We found that the suggested ordering introduced some structural difficulties– specifically we wanted to keep ventilation and ADC together both because of their interplay at the diffusion front and to avoid some of the back-and-forth between gas and dissolved that the reviewer found confusing. We also find value in keeping the ‘practical imaging’ section in one piece– even if optimal strategies do differ somewhat, there are sampling schemes, T2*- and non-renewable-magnetization-ratelated challenges, etc. that are common to all HXe imaging. We intended this section to explain these challenges and how people have approached them, all in one place, to explicitly facilitate that kind of comparison in a way that is often difficult in treatments with either a more clinical or a more basic physics focus.
- What surprised me, is the complete absence of references in the introduction section. There are many statements in the introduction that should be referenced properly. For example, statements such as “Among imaging modalities, MRI is particularly well suited to this task because it derives from and can make use of NMR-based techniques that encode rapid dynamical behavior as changes in the frequency, linewidth or intensity of spectral features.”, “Lung disease and its diagnosis is therefore extremely challenging, requiring an understanding of the muscular and airflow aspects of breathing on the scale of seconds and centimeters, to the diffusive dynamics of gases in the sub-millimeter regime, down to the millisecond- and micron-scale transport of gases across capillary walls and in red blood cells.”, and “This includes 1H metabolite or fat/water imaging, or hyperpolarized 13C metabolic imaging.” should be referenced. The authors should go over the introduction and include proper citations throughout.
We have added eighteen references to the introduction as the reviewer suggests.
- Some images seem to be low resolution and blurry. The authors should improve the visibility of the figures. Specifically, figures 2.4, 2.5, and 5.3.
Done.
- In the Whole Body and Brain Imaging section, the authors should include recent preclinical achievements in HP 129Xe molecular imaging (results by McHugh CT., et. al. Magn Reson Imag. 2021; 87(3); 1480-1489 as well as Hane FT., et.al. Sci Rep. 2017; 7:41027). This field is unique for HP 129Xe MRI and recent preclinical success in hyperpolarized chemical exchange saturation transfer is of high interest for HP 129Xe MRI and molecular imaging.
A brief discussion of HyperCEST (section 7.5) with special attention to the two articles mentioned has been added to the Whole Body and Brian Imaging section:
A novel modality for dissolved-phase imaging that has only recently been demonstrated in-vivo123 is hyperpolarized chemical exchange saturation transfer (HyperCEST) imaging. Chemical exchange saturation transfer (CEST) is a molecular imaging technique that selectively excites a solute proton of interest and then measures the transfer of saturation between the excited molecule and the bulk water protons in the body. HyperCEST is an analogous technique that uses HXe signal in lieu of the 1H signal. Specifically, the hyperCEST technique involves saturating 129Xe atoms caged in supramolecular hosts and then measuring the change in signal as the caged atoms exchange with HXe dissolved-phase atoms outside of the molecule. Although there still exists significant practical challenges while implementing hyperCEST in vivo124, it has the potential to provide far more sensitive imaging measurements that traditional dissolved-phase HXe imaging.
Specific comments
- Page 2, Introduction. “…breathing, heart rate and circulatory rates in rodent models are far higher, as are the available imaging magnetic fields and gradients.”
It should be mentioned as well that preclinical HP 129Xe imaging is also associated with technical challenges such as a requirement for the small dedicated coils, additional equipment such as an animal breathing ventilator, etc.
Done.
- Page4. Section 2. Eqns. 1-2. Some dots appear in the equations at random places. For example, in Eqn. 2 a dot appeared on top of both minus signs. Eqn.1. has a dot on top of V in nominator in the first term. Is that a different volume than in the denominator? Please correct or explain the meaning of those dots.
We apologize for the confusion, the dots represented time derivatives and we failed to notice that some of them moved around in the equation editor. We have replaced these by the more familiar dX/dt.
- Page 6. Section 3. While talking about the implementation of ADC gradients into spectroscopy, the authors may also reference work by Boudreau M., et. al. (Boudreau M., et. al. Magn Reson Med. 2013;69(1):211-220).
Done as suggested.
- Page 11. Section 4. “Within approximately 100ms after inhalation, dissolved xenon accounts for ~0.2-4% of the total 129Xe signal, depending on species and disease state.”
The author should provide a reference for this statement.
Done as suggested.
- Page 11. Section 4. “The limited amount of dissolved xenon, along with the short T2* characteristic of lung tissue …”.
The authors should provide a T2* value for HP 129Xe dissolved in the lung parenchyma.
Done.
- Page 12. Section 4. Figure 4.5 looks like three figures of different sizes rather than a single figure. The authors should make sure, that plots A, B, and C are actually united into a single graphical object.
Done.
- Page 12, Section 4. “…these peaks appear over a much longer timescale, as shown in Figure 3.5A.”.I guess the authors meant Figure 4.5 here. Please double-check., 8. Page 13. Section 4. “…spectroscopy, as shown in Figure 3.5B”. Again, I assume it should be Figure 4.5B
Corrected.
- It seems that there is no reference to Figure 4.5C in the text. If there is no reference to it, the authors should remove Figure 4.5C from the manuscript. Alternatively, please reference Figure 4.5C where it is appropriate.
Reference added in the text.
- Page 15. Section 5.2.1. Was natural abundance (~26%) used for the calculation of HP 129Xe magnetization or isotopic enrichment of 129Xe (~85-90%) was assumed? Please specify.
We assumed 85% enrichment and 80% alveolar xenon gas (i.e., 80%/20% HXe/O2 washed in over multiple breaths) as specified in the table caption.
- Page 16. Section 5.2.1. Table 5.4. The authors should change the table to reflect polarization values used from different polarizers in different groups. The polarization column should reflect a practically achievable polarization range. For example, 15% polarization was used for preclinical imaging by McHugh CT., et.al. (McHugh CT., et. al. Magn Reson Imag. 2021; 87(3); 1480-1489), whereas the highest polarization reported for animal scans was ~30% (Chahal S., et. al. Brain imaging using hyperpolarized 129 Xe magnetic resonance imaging. In: Eckenhoff RG, Dmochowski IJ, eds. Methods in Enzymology. Vol 603. Academic Press Inc. 2018:305-320.). The obtained magnetization range for the 0.15-0.3 polarization range should be provided in the table (with appropriate references).
The reviewer is correct, in that the magnetization does vary among publications, groups, and commercial or home-made polarization equipment. We intended this table to give a rough idea of how magnetization compares between gas and dissolved, and between gas and 1H lung tissue, in order to give the reader a sense of the general scale of the available signal, not to provide a complete survey of who is working with what sort of signal levels. We feel that would distract from the simple message of the table and have therefore kept the same overall format.
- Page 16. Section 5.2.2. Figure 5.5. I suggest including the image acquired using the shown UTE3D pulse sequence in the Figure. This will allow the reader to understand the outcome image right away.
We agree with the reviewer that including the image will be beneficial to the reader. We have now included the image within Figure 5.5, and updated the text to reference it.
“These considerations are exemplified in Figure 8.5, which shows the UTE3D pulse sequence, time domain FID data for gas-phase xenon MRI and the resulting image in a mechanically ventilated rat.”
- Page 16. Section 5.2.2. “… the transverse magnetization decays with a short T2* of ~5ms at 7 Tesla49”. I suggest writing 7T to maintain consistency with the other manuscript sections.
Done.
- Page 16. Section 5.2.2. “…as GRE, 3D radial (UTE3D), spiral,…”
The authors should include references for each technique mentioned.
We have now included relevant references for each technique
- Page 17. Section 5.2.2. “Typically, the scan time is limited to ~10 minutes…”
We have updated the text as below and included relevant references
“Typically the scan time is limited to <20 minutes to limit substantial differences in the signal from T1 relaxation over the duration of the scan (ref)”
- Page 17. Section 5.2.2. “Most commonly, a fixed small flip angle is used, which induces a modest amount of depolarization with each excitation.”
References are required for this statement.
In an effort to be more objective, we have revised this statement and included more references of implementations of fixed and variable flip angle strategies.
“A fixed small flip angle can be used for every excitation, which induces a modest amount of depolarization with each excitation (refs). Alternatively, a variable flip angle strategy can be employed (refs)...”
- Page 18. Section 5.2.4. “…technique, where a single echo time is employed, which corresponds to the two resonances being 90° out of phase…”.
The authors should include the value for TE.
We have included reported values of TE90 at 2T and 7T and added references:
“The simplest implementation is the 1-point Dixon technique, where a single echo time is employed, which corresponds to the two resonances being 90° out of phase with each other (860-940 μs in rats at 2T (ref); 248 ± 5 μs in rats at 7T (ref)).”
- Page 22. Section 5.3. Table 5.1. The authors gathered eleven studies in the table and mentioned them in the text. However, the main findings of those works were not highlighted. Based on Table 6.1 it seems that septal wall thickness and hematocrit are two of the most commonly used biological parameters for gas exchange studies. The author should at least describe disease-caused changes in these parameters. Providing the explanation along with equations for septal wall thickness and hematocrit calculation would be also helpful (and useful) for the reader.
We have added two paragraphs to section 5.1 highlighting septal thickness and haematocrit:
“As shown in Table 5.1, septal wall thickness and haematocrit are the two most commonly studied measures with these models and, so, it is worth expounding on their formulation and physiological relevance. The alveolar septum is a thin wall that separates adjacent alveoli. The thickness of the alveolar septum has two effects on the CSSR uptake curve. A thicker alveolar septum limits the rate at which 129Xe gas can dissolve into the blood and thus affects the initial slope of the uptake curve. Secondly, the finite thickness of the septum affects the saturation capacity of the lung tissue which is presented by the latter flat portion of the CSSR curve. Septal thickness can be affected by various lung conditions including interstitial lung disease and radiation induced lung injury.
Haematocrit is the percentage volume of red blood cells in the blood and, thus, affects the oxygen (and 129Xe) carrying capacity of the blood. Hence, haematocrit is calculated using the ratio of barrier to red blood cell signals (and cannot be calculated from the Patz model that considers both compartments together). Low haematocrit is a sign of anemia and can be caused by cancers such as leukemia or by other conditions such as radiation induced lung injury.”
- Page 30. Section 8.3. The author mentioned Peled’s model only, which was historically the first model of the HP 129Xe signal in the brain. The authors should also talk about and discuss the other HP 129Xe uptake models such as Martin’s model (Martin CC., et. al. J Magn Reson Imag. 1997; 7:848-854), Kimura’s model (Kimura A., et. al. Magn Reson Med Sci. 2008; 7:179-185), and Imai’s model (Imai H., et. al. NMR Biomed. 2012; 25:210-217). I would encourage the authors to compare these models similarly to how they compared the models of HP 129Xe gas exchange in the lungs.
In the first paragraph of section 7.3, the following line was added:
“Later, Kimura et al and Imai et al formulated similar equations based on the same principles but accounting for the reduction in hyperpolarization due to RF excitations110,111.”
In addition, the following paragraph regarding Martin’s model was added to the end of section 7.3:
“Although the Peled model has been the predominant basis for 129Xe uptake models used in the field, Martin et al published a similar but different formulation around the same time122. While the Peled model began with a 129Xe reservoir in the lungs, their model included transport of 129Xe to the lungs via the mouth and throat and, therefore, can accommodate a variety of breathing maneuvers.”
Round 2
Reviewer 2 Report
I would like to thank the authors for addressing all my comments. I believe the manuscript can be accepted now.